# Multi-scale dynamic imaging reveals that cooperative motility behaviors promote efficient predation in bacteria

Sara Rombouts[1], Anna Mas [1], Antoine Le Gall[1] ✉, Jean-Bernard Fiche[1], Tâm Mignot[2] & Marcelo Nollmann [1] ✉

Many species, such as fish schools or bird flocks, rely on collective motion to forage, prey, or escape predators. Likewise, *Myxococcus xanthus* forages and moves collectively to prey and feed on other bacterial species. These activities require two distinct motility machines enabling adventurous (A) and social (S) gliding, however when and how these mechanisms are used has remained elusive. Here, we address this long-standing question by applying multiscale semantic cell tracking during predation. We show that: (1) foragers and swarms can comprise A- and S-motile cells, with single cells exchanging frequently between these groups; (2) A-motility is critical to ensure the directional movement of both foragers and swarms; (3) the combined action of A- and S-motile cells within swarms leads to increased predation efficiencies. These results challenge the notion that A- and S-motilities are exclusive to foragers and swarms, and show that these machines act synergistically to enhance predation efficiency.

Collective movement is employed by many organisms, including fish, birds, and ants, for the rapid exploration, killing, and predation of local resources[1]. Remarkably, multicellular systems also display collective cell movement to achieve similar goals. For instance, neutrophils swarming to kill invading microorganisms[2–4], amoeba feeding on bacteria[5], and bacterial killing and predation within the gut microbiota[6] or within natural ecosystems[7].

*Myxococcus xanthus* is a social bacterium that assembles multicellular biofilms to collectively hunt and attack other microorganisms, including bacteria, fungi, and yeast[7–10]. To achieve this, *M. xanthus* cells glide over solid surfaces by two genetically independent motility mechanisms[8,11]. Social (S-) motility pulls cells forward by extending and retracting Type IV pili[12,13], whereas Adventurous (A-) gliding assembles a multicomponent focal adhesion engine powered by proton-motive force to propel the cell[14,15]. Notably, A- and S-motility mediate distinct and complementary tasks: A-motility in driving the movement of single cells at the colony edges ([8,11];[7–10]), and S-motility in promoting the coordinated movement of cells as large multicellular groups (swarms)[9,10,16]. However, beyond these observations, it was not clear how each system exactly contributes to the *Myxococcus* lifestyle; what are the specific roles and added values of these motility systems. Recently, A-motility and contact-dependent killing were shown to be necessary for prey colony penetration, suggesting that multiple motility systems are required for efficient predation. However, the exact function of S-motility was not determined[17].

Here, we investigate the roles of each of these motility mechanisms during *M. xanthus* predation by applying a high-throughput method to track single prey and predator cells in dense biofilms over extended periods, with high temporal and spatial resolutions.

## Results

### Monitoring dynamics of predation at single-cell resolution

To study the dynamics of bacterial predation with single-cell resolution, we implemented a time-resolved version of bactoHubble, an imaging-based method that enables visualization of whole bacterial communities with single-cell resolution[18]. For this, we built a robotized microscope able to acquire 3D, multiple-color images of large areas at diffraction-limited resolutions for long time periods (~hours; Fig. 1a)[19].

[1]Centre de Biologie Structurale, CNRS UMR 5048, INSERM U1054, Université de Montpellier, 60 rue de Navacelles, 34090 Montpellier, France. [2]Laboratoire de Chimie Bactérienne, Marseille, France. ✉e-mail: antoine.legall@cbs.cnrs.fr; nollmann@cbs.cnrs.fr

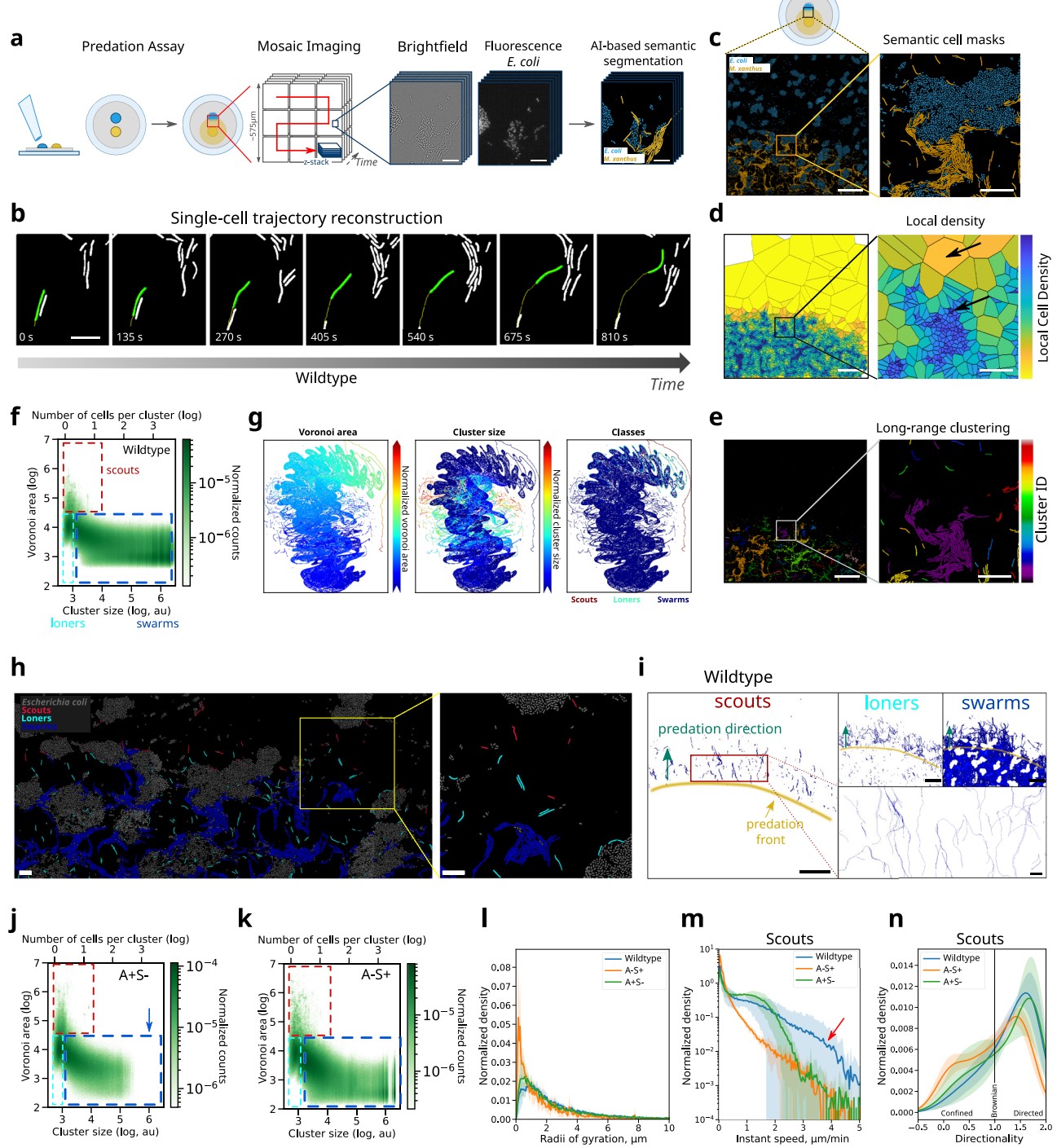

Three-dimensional acquisitions enabled correction of axial drift and compensation of axial deformations in the substrate by adaptive reprojection, to produce in-focus 2D images (see "Methods" section). To cope with the high acquisition throughput —over 10,000 images per experiment— we implemented a deep learning automatic semantic segmentation approach coupled to an algorithm that links cell masks at different times to enable the reconstruction of single-cell trajectories in 2D over long time periods (~7 h; Fig. 1b; see "Methods" section). To capture the dynamics of *M. xanthus* predation at single-cell resolution, we adapted the classical predation assay of *E. coli* by *M. xanthus*[18,20] where these species interact over a 1 cm² agar surface directly on a microscope slide (see Predation assay in "Methods" section). In these conditions, the invasion process occurs over a single

prey cell layer, allowing identification of single predator and prey cells at any given stage.

Next, we used a two-pronged approach to retrieve spatial organization information from the semantic, single-cell segmentation of prey and predator (Fig. 1c). First, we produced a Voronoi tessellation based on the middle points of the backbones of each *M. xanthus* mask (Fig. 1d, see Voronoi tessellation in "Methods" section). Thus, the area of the polygon associated with each cell mask provided a proxy for local cell density: *M. xanthus* cells associated with large Voronoi areas were relatively isolated from other *M. xanthus* cells, whereas cells associated with small Voronoi areas were located in high cell density regions (Fig. 1d, arrows). Second, we partitioned groups of *M. xanthus* cells into clusters by linking together cells located in close spatial

**Fig. 1 | Multiscale dynamic imaging reveals four distinct cell classes. a** Schematic of large-scale imaging and semantic segmentation of the predation zone. Scalebars = 10 μm. **b** Example of a single-cell trajectory (yellow line) reconstructed by connecting segmented cell masks of the same cell (green mask) over time. Scalebar = 5 μm. **c** Example of a semantically segmented large ROI at the predation front containing the masks for *M. xanthus* (yellow) and *E. coli* (blue) (left) Scalebar = 100 μm. The zoom of the boxed area shows single segmented cells in high and low cell density areas (right). Scalebar = 20 μm. **d** Voronoi tessellation of a large ROI at the predation forefront calculated from the middle points of the *M. xanthus* mask backbones (left) Scalebar = 100 μm. The zoom of the boxed area shows small polygon areas for *M. xanthus* cells in high *M. xanthus* cell density regions and large polygon areas for cells in low *M. xanthus* cell density regions (right). Black arrows point to examples of isolated and densely packed cells. Scalebar = 20 μm. **e** Long-range clustering of *M. xanthus* cells in close spatial proximity in a large ROI at the predation forefront (left). Scalebar = 100 μm. The zoom of the boxed area shows several cell clusters of various sizes (right) Scalebar = 20 μm. **f** 2D Voronoi area-cluster size histogram for wild-type data accumulated from six experimental replicates (see Fig. S1a for single replicated examples). Red, cyan, and blue boxed areas correspond to scout, loner, and swarm cell classes, respectively.
**g** Dimensionality reduction analysis of cluster size and Voronoi area parameters with UMAP. Left, middle, and right panels represent aggregated single cells data points color-coded with cluster size, Voronoi area values or classes as defined by our criteria, respectively. **h** Examples of single cells masks color-coded with cell classes in wild-type conditions. *Escherichia coli* cell masks are grey, *M. xanthus* scouts, loners, and swarms are colored in red, cyan, and blue, respectively. Scale bar

is 10 μm. **i** Spatial occupation of scout, loner and swarm trajectories for wild type at the predation forefront. The yellow line delimitates the predation front, the green arrow indicates the direction in which *M. xanthus* predator cells move through the prey colony (predation direction). Scalebars = 100 μm. Magnified image on the bottom right shows long scouts trajectories (Scalebar = 15 μm). **j** 2D Voronoi area-cluster size histogram for A-motile cells (A+S-) data accumulated from four experimental replicates (see Fig. S1a for single replicate examples). Blue arrow highlights the absence of density for large swarms. **k** 2D Voronoi area-cluster size histogram for S-motile cells (A-S+) data accumulated from four experimental replicates (see Fig. S1a for single replicate examples). **l** Histogram of gyration radii of scouts cells trajectories in wild type, A-motile (A+S-) and S-motile (A-S+) communities. Shaded areas highlight the standard deviations from the mean (solid line) of six experimental replicates for the wild-type and four replicates for each mutant strains. **m** Histogram of instantaneous speed of scouts cells in wild type, A-motile (A+S-) and S-motile (A-S+) communities. Shaded areas highlight the standard deviations from the mean (solid line) of six experimental replicates for the wild-type and four replicates for each mutant strains (see Fig. S1g for single replicate examples). Red arrow points to speed values only reached by wild-type cells with both motility systems. **n** Histogram of movement directionality of scouts cells in wild type, A-motile (A+S-) and S-motile (A-S+) communities with directionality lower than 1 being confined motion, equal to 1 being Brownian motion and larger than 1 being directed motion. Shaded areas highlight the standard deviations from the mean (solid line) of six experimental replicates for the wild-type and four replicates for each mutant strains (see Fig. S1h for single replicate examples).

proximity (Fig. 1e, see long-range clustering in "Methods" section), and calculate the number of cells in each cluster and its size. These cluster statistics were associated with Voronoi local density measurements to quantify the local environment of each single *M. xanthus* cell at each specific time during predation.

By representing each *M. xanthus* cell by its local density and the size of the associated cell cluster, we observed that single cells tend to scatter along an L-shaped, continuous distribution (Fig. 1f and S1a, see 2D histograms in "Methods" section). From this distribution, we defined three cell classes (Fig. 1f, see classes in "Methods" section). To verify the validity of this classification, we turned to Uniform Manifold Approximation and Projection for Dimension Reduction (UMAP) (see UMAP Projection in "Methods" section). First, we tested whether local density or cluster size were sufficient to label cells that were segregated in the UMAP space (Fig. 1g, left and center panel). In fact, neither of these single parameters was enough as both displayed a continuous change within the UMAP space. Next, we color-coded each single cell as either scout, loner or swarm according to our classification that made use of both local density and cluster size information. In this case, we observed that single scout cells clearly segregated from loners and swarms in the UMAP (Fig. 1g, right panel). The spatial segregation between loner and swarm cells was incomplete, but loner cells still occupied a distinct region of the map. The partial overlap between these two classes is not surprising, as small groups of swarm cells splitting from a swarm would have similar properties (local density, cluster size) as loners. The three classes are: (1) scouts which represent small groups of cells (1-20) isolated from the main colony, typically localizing ahead of the forefront of the invading wave; (2) swarm cells which lie within cell clusters and are always closely packed with other cells; and (3) loners cells which lie close to other cells within the colony front (see snapshots and trajectories in Fig. 1h, i). We note that while scouts segregated into a well-defined cluster in the UMAP representation (Fig. 1g), loners and swarms displayed a partial overlap. Nonetheless, changes in the analysis and classification thresholds had moderate effects on the classification results (Fig. S1b-d). The full significance of this classification emerges when the behaviors of A- or S- mutants during *E. coli* predation are explored.

Loss of S-motility (A+S-) led to colonies with reduced swarm sizes at the invasion front (Fig. 1j, blue arrow and Fig. S1e), consistent with the classical result of Hodgkin[8]. As expected, scouts and loners were

still present in this community (Fig. 1j, red and cyan boxes). In contrast, loss of A-motility (A-S+) produced communities with large swarms, but also with scouts and loners (Figs. 1k and S1a,e). This result seems in contrast to the classical view of A-motility being required to produce isolated single cells in non-predating conditions[9]. To understand this apparent discrepancy, we compared scout trajectories in wild-type and A-S+ communities. We observed that scouts in A-S+ colonies traveled considerably shorter distances than wild-type (Figs. 1l and S1f), which is expected since Type IV pili allow movement of cells only if attachment prey cells are within range[21,22]. Only comparing populations of motile cells yielded classes consistent with the view that A-motility is associated with exploratory behaviors of single cells (Fig. S1e). Next, we analyzed the distributions of instantaneous speeds and overall directionality of movement for scouts in these three conditions (see calculation of speeds and mean squared displacement calculation in "Methods" section). Notably, A-S+ scouts displayed a marked reduction in speed (Figs. 1m and S1g), and a reduction in directed motion counterbalanced by a gain in Brownian and confined movements (Figs. 1n and S1h). This result is in line with the finding that during colony expansion single, isolated *M. xanthus* cells (i.e. scouts/loners) move only if they carry a complete A-motility system[8]. We also note that wild-type scouts reach higher speeds than either A-S+ or A+S- scouts (Fig. 1m, red arrow). Differences in speed and directionality between wild type and A-S+ conditions were found statistically significant (Fig. S1i, j) and distributions were found robust to the classification parameters (Fig. S1k–n). All in all, these findings suggest that A-motility is required to observe motile scouts, however presence of both motility systems synergistically enhances the speed of wild-type scouts.

### Cells frequently transit from individual to collective states

Next, we explored whether motile cells were able to dynamically transition between cell classes during predation. For this, we calculated the Voronoi area and number of cells in a cell cluster to assign classes following the criteria established above (Fig. 1f). Then we embedded single-cell trajectories in a space where each cell class was assigned to a corner of a triangle, thus transitions appeared as the edges of the triangle (Fig. 2a, see transition analysis in "Methods" section). Transitions between cell classes occurred very frequently as many cells changed classes at least once during the time course of the

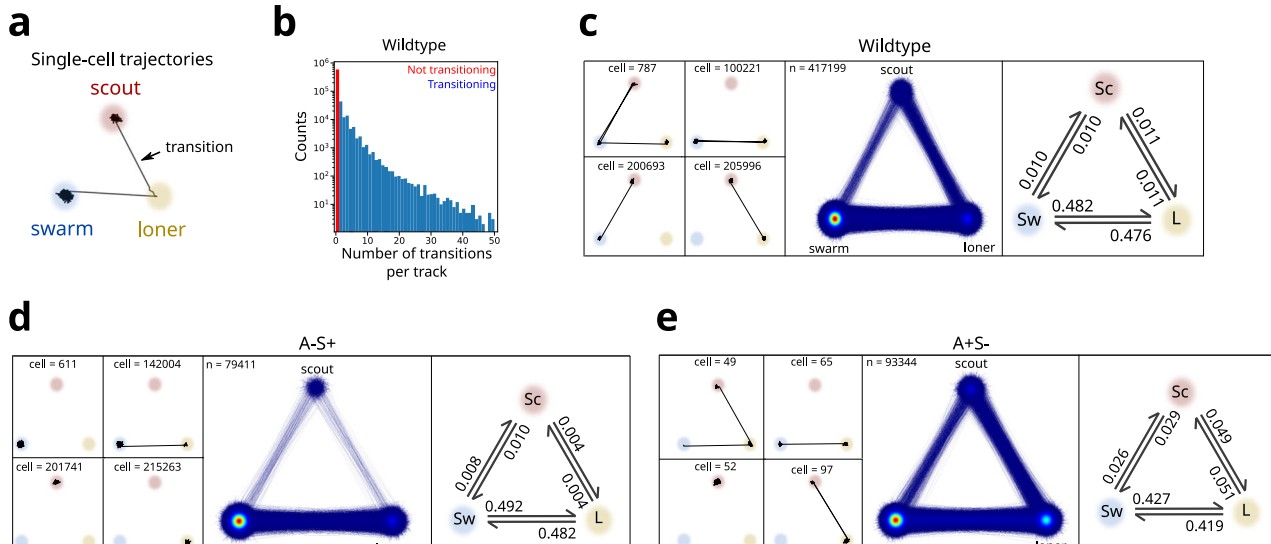

**Fig. 2 | Cells frequently transit from individual to collective states. a** Schematic line plot of cell class transitions observed in a single-cell trajectory. **b** Histogram of number of motile cells transitions per trajectory for wild-type data accumulated from six experimental replicates. **c** Single-cell transitions of motile cells for wild-type data accumulated from six experimental replicates. Left: examples of state transitions occurring in single-cell trajectories of wild-type cells. Track IDs are indicated in each example. Middle: overlay of state transitions from all trajectories. Right: state transition probabilities between classes. "*n*" represents the total number of tracks. **d**, **e** Same representations as in **c** but for S-motile (A-S+) cells (**d**) and A-motile (A+S-) cells (**e**).

experiment (Fig. 2b). These state transitions could be readily detected in single-cell trajectories (Fig. 2c, left panels). To determine which transitions were most common, we overlapped thousands of single-cell trajectories within the same diagram (Fig. 2c, middle panel). Notably, transitions between classes were very common, and the most common route to becoming scouts typically involved cells in swarms transitioning through a lone state (Fig. 2c, middle panel).

To quantify the relative transition frequencies and to determine whether transitions were symmetric, we calculated the state transition probabilities (Fig. 2c, right panel). Remarkably, forward and reverse transitions between states were equally probable, ensuring constant steady-state populations over our acquisition time. Importantly, state transitions were also frequent and symmetric in communities lacking A- or S-motility (Figs. 2d, e and S2a–c), thus the ability of cells to transition back and forth between states did not seem to depend on either motility system. The kinetic rates to and from scout cells in A-S+ communities were lower than for wild-type, indicating that these transitions are enhanced by A-motile cells. This is consistent with more frequent transitions to and from scouts in A+S- cells. Overall, these results show that transitions between cell classes are frequent and bi-directional, enabling cells to dynamically change their role within the community and ensuring that cell groups remain populated during the advancement of the predation wave. These findings prompted us to investigate whether A- and S-motile cells strictly segregate according to specific cell classes (i.e. scouts and swarms), given that A- and S-motilities have been previously associated with the movement of scouts and swarms, respectively.

### A-motile cells are present in all cell classes and are required to drive collective cell movement

To first address this issue, we imaged AglZ, an integral component of the Agl-Glt machinery that assembles as polar clusters in A-motile cells[14,15] in wild-type *M. xanthus*. As expected, small groups of cells often displayed polar AglZ clusters (Fig. 3a, green circles, see AglZ foci detection in "Methods" section), consistent with our result that scout movement requires A-motility (Fig. 1l, m). Surprisingly, however, large groups of cells also frequently displayed polar AglZ clusters (Fig. 3b, green circles). Thus, these results suggest that both A- and S-motile cells may be present within scouts and swarm groups.

To further test this hypothesis, we simultaneously imaged A- and S-motile cells within scout and swarm groups during predation. For this, we mixed cultures of mCherry-labeled A+S- cells with GFP-labeled A-S+ cells, spotted them together, and imaged them using two-color microscopy (see Predation assays in "Methods" section). Both motility mutants moved together away from the spotting site to reach the prey, indicating that both A+S- and A-S+ cells travel together over long distances (Fig. 3c). Notably, A+S- and A-S+ cells thoroughly intermingled both in small (i.e. loners, scouts) and large cell groups (i.e. swarms; Fig. 3c). Thus, these observations indicate that S-motile cells can often intermingle with A-motile cells in scout groups, and that A-motility may not only be used by isolated cells away from the forefront, but may also play a role in collective cell movements.

We tested these hypotheses by a multi-pronged approach. First, we monitored whether A-motility and S-motility mutants induced changes in motile cell populations by subtracting the density/cluster-size histograms of pure cultures of A-S+ and A+S- communities by that of wild-type communities (Fig. 3d, e, see Two-dimensional histograms in "Methods" section). Interestingly, the removal of A-motility led to an overall reduction in the accessible Voronoi area for both scouts and swarms (Fig. 3d), thus collective cell groups tend to remain closer to one another in absence of A-motility. Conversely, upon removal of S-motility, both individual and collective cell groups dispersed to occupy larger accessible Voronoi areas away from the community forefront, while cohesive populations decreased in density (Fig. 3e). These results, therefore, support a model where A-motility is associated with exploratory behaviors in scouts and swarms while S-motility is associated with the cohesion of cell groups.

Secondly, we investigated the functional role of A-motility in the movement of swarms by quantifying their instantaneous speed distributions in wild-type and mutant communities. Remarkably, the removal of A-motility led to a large decrease in instantaneous swarm speeds (Figs. 3f and S3a). Interestingly, removal of S-motility only had an effect on the high instantaneous speeds of cells within swarms, suggesting that A-motility is sufficient to drive the movement of collective groups but may be synergistically enhanced when combined with S-motility. Differences in speed between wild type and A-S+ conditions were found statistically significant (Fig. S3b) and swarm speed distributions were found robust to the classification parameters

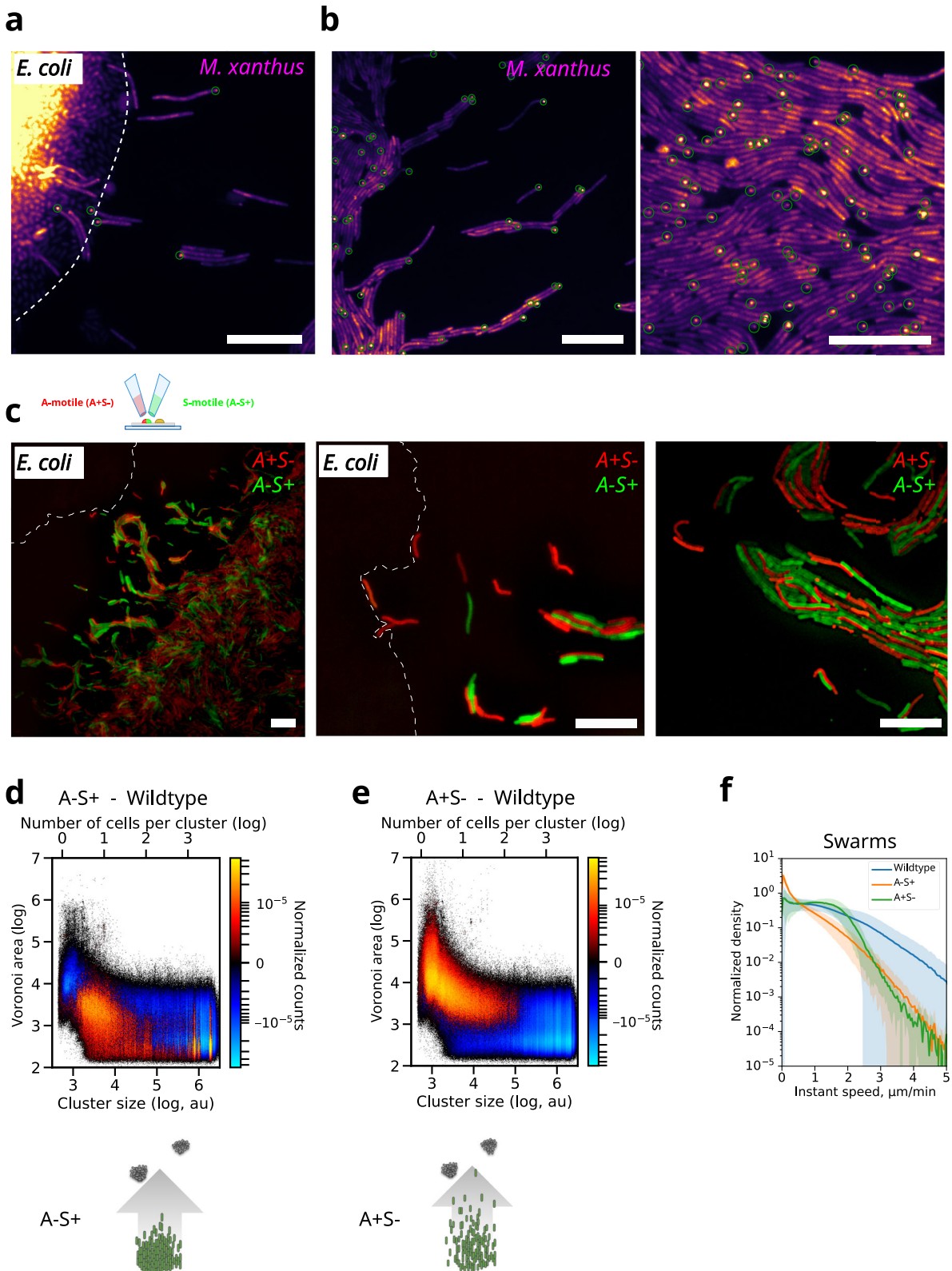

(Fig. S3c, d). All in all, these results suggest that A-motile cells inter-mingle with S-motile cells and play a functional role in the collective movement of all cell populations.

### A-motile cells drive the ability of multicellular groups to follow trails deposited by scouts

To study the role of A-motility in the directional movement of collective cell groups, we analyzed whether swarms followed the movement of scouts during colony invasion. Strikingly, the path taken by scouts was followed by other cells, and increased in width as larger groups of cells passed over it (Fig. 4a). To quantify this behavior, we segmented the trails left by scouts (Figs. 4b and S4a, left) and those left by swarms (Figs. 4b and S4a, middle) ahead of the predation front, and calculated their similarity index (SI) map (see similarity index map in "Methods" section). The SI map is close to zero where scouts and swarms follow different paths and close to unity where their paths overlap spatially.

**Fig. 3 | A-motile cells mix with S-motile cells in all population classes.**
**a**, **b** Fluorescence images of A-motility complexes (WT AglZ-NeonGreen) in isolated cells (a) or in groups of cells (b) at the predation front. Green circles highlight automatically detected AglZ-clusters in single cells. Two experiments were repeated independently with similar results. Scalebar = 10 μm. **c** Left panel: migration of a 50/50 mixed community of A-motile (A+S-) (outer-membrane-mCherry, red) and S-motile (A-S+) (outer-membrane-sfGFP, green) cells at the predation front towards *E. coli* micro-colonies (top left, bright field). Scalebar = 10 μm. Middle and right panels: zoomed-in images of a 50/50 mixed community of A-motile (A+S-) and S-motile (A-S+) showing isolated A- and S- motility cells (left) and A- and S- motility

collective groups (right). Three experiments were repeated independently with similar results. **d**, **e** Difference of A-S+ (Fig. 3d) and A+S- (Fig. 3e) with wild-type normalized 2D Voronoi area-cluster size histograms of motile cells. To account for motile cells only, histograms of Fig. 1f, j, k were thresholded for normalized distance higher than 2 pixels/track length. Schemes depict the distribution of *M. xanthus* cells (green) towards *E. coli* (gray). **f** Histograms of instantaneous speed of swarm cells in wild type, A-motile (A+S-) and S-motile (A-S+) communities. Shaded areas highlight the standard deviations from the mean (solid line) of six experimental replicates for the wild-type and four replicates for each mutant strains (see Fig. S3a for single replicate examples).

Notably, we observed that swarms tended to follow the trails left by scouts over the whole predation front (Figs. 4b and S4a).

Next, we tested the role of A-motility in this process by exploring the ability of swarms to follow scouts in an A-S+ community within the active exploration region. Notably, the number and length of overlapping trails were dramatically reduced in these communities (Figs. 4c and S4b). In contrast, A+S- cells displayed similar trails to wild-type (Figs. 4d and S4c). These results can be quantified by calculating the histogram of similarity track sizes for these three communities (Figs. 4e and S4d). This analysis clearly showed a drastic decrease in the number of common trails for all trail lengths for the A-S+ community. All in all, these results show that A-motility is required for the ability of swarms to follow the trails of scouts, consistent with previous reports[23–26].

To investigate whether swarms were able to follow trails within the active predation region behind the predation front, we built a map where single-cell trajectories of all cell groups were overlapped and color-coded by time (Fig. 4f and S4e, see trail maps in "Methods" section). In these maps, trails are represented by regions displaying overlapping tracks of different colors. As in our previous analysis, trails are observed not only within the exploration zone but also within the predation region, occupied primarily by swarms. From these analyses, we conclude that trails left by scouts are widely followed by swarms during prey invasion and predation.

To determine whether formation of trails within the predation region requires A- or S-motilities, we constructed time-colored trail maps for A-S+ and A+S- communities (Figs. 4g, h and S4f, g). Notably, tracks did not overlap within the predation zone in A-S+ communities, consistent with the formation and use of trails within the predation zone requiring A-motility. In contrast, we observed overlapping tracks in A+S- communities that appeared narrower than in wild-type conditions, likely due to the reduction of swarm group sizes. We quantified these observations by calculating the distribution of swarm directionalities for wild-type, A-S+ and A+S- communities (Figs. 4i and S4h), and observed that loss of A-motility led to less directed and more confined/Brownian motion. Differences in directionality between wild-type and A-S+ conditions were found statistically significant (Fig. S4i) and directionality contributions were robust to the classification parameters (Fig. S4j, k). Overall, these results indicate that A-motility is required to lay trails and to promote the directional movement of multicellular groups at the community front, while S-motility plays a role in strengthening their cohesion. However, it is unclear whether and how the presence of both motility mechanisms contributes to efficient predation, which could explain why these two systems are concurrently maintained during evolution.

### Efficient predation requires the action of A- and S-motile cells

To address this issue, we monitored prey lysis during the predation process by exploiting the ability of our method to perform semantic segmentation over large areas and extended time periods. From the semantic segmentation, we defined the 'invaded zone' as the region occupied partially or totally by *M. xanthus* during our acquisition time, and the 'safe zone' as the region not yet reached by *M. xanthus* cells (Fig. 5a, right panel). Inspection of the raw fluorescence images of *E.*

*coli* further revealed that effective prey lysis does not occur homogeneously over the invaded zone, but is more pronounced behind the invasion front (hereafter 'predation zone', see Fig. 5b, right panel). To quantify this observation, we computed the ratio map between the last and the first images, using both the *E. coli* and *M. xanthus* raw fluorescence images (Fig. 5c, d). The *E. coli* map exhibits a large decrease in E. *coli* fluorescence signal within the invaded zone (Fig. 5c), while the *M. xanthus* map clearly shows the progression of the scouts and swarms fronts over the course of the acquisition (Fig. 5d). Notably, prey lysis is low within the scouting front, but is instead severe within the region occupied by swarms (Fig. 5d right panel). These observations suggest that swarms may play a key role in prey lysis.

To obtain a more accurate estimation of prey lysis, we quantified the integrated fluorescence signal of the prey within the predation and safe zones over time (Fig. 5e, see prey consumption in "Methods" section). The integrated normalized prey fluorescence decreased slightly and monotonically over time within the safe zone due to photobleaching. Notably, the reduction in total normalized fluorescence was dramatically faster within the predation zone where most prey was lysed (Fig. 5e). We note that while wild-type communities lyse the overwhelming majority of prey (Fig. 5b), the relative degree of lysis can depend on the densities and spatial distributions of *M. xanthus* and *E. coli*.

To investigate the roles of A- and S- motility in predation, we performed similar analysis and quantification for A-S+ and A+S- communities (Fig. S5a–d). The absence of A-motile cells led to a marked reduction in the predation ability of *M. xanthus* (Fig. 5f, h), likely linked to the reduced ability of this community to move directionally, efficiently explore ahead of the predation zone, and form and follow trails. We note that the requirement of A-motility could be more pronounced depending on the density of the prey. In conditions when *E. coli* was grown to confluence, A-motility was essential for prey penetration[17]. Notably, we observed that A+S- communities lyse prey cells at a rate considerably lower than either wild-type or A-S+ communities (Fig. 5g, h). This suggests that S-motile cells within swarms contribute to the predation efficiency of wild-type communities, presumably by the principle of mass action. These results highlight the need for both A- and S-motility systems for efficient predation.

## Discussion

The ability of *M. xanthus* to glide on solid surfaces relies on two distinct and genetically independent molecular machines that could be alternatively used to adapt to the mechanical properties of the substrate[27]. Since the discovery and first microscopic characterization of *M. xanthus* movement, A- and S-motile cells were thought to segregate spatially and to behave in two distinct manners: A-motile cells moving in isolation ahead of the invasion front acting as foragers searching for nutrients, and S-motile cells assembling large collective cell groups at the rear by promoting social interactions[9]. In contrast, our data show that S-motility mutants (A-motile cells) and A-motility mutants (S-motile cells) intermingle together within collective cell groups. Thus, A- and S-motile cells are not necessarily spatially segregated during predation, challenging the current view that A-motility is solely attributed to the movement of individual cells and S-motility to the

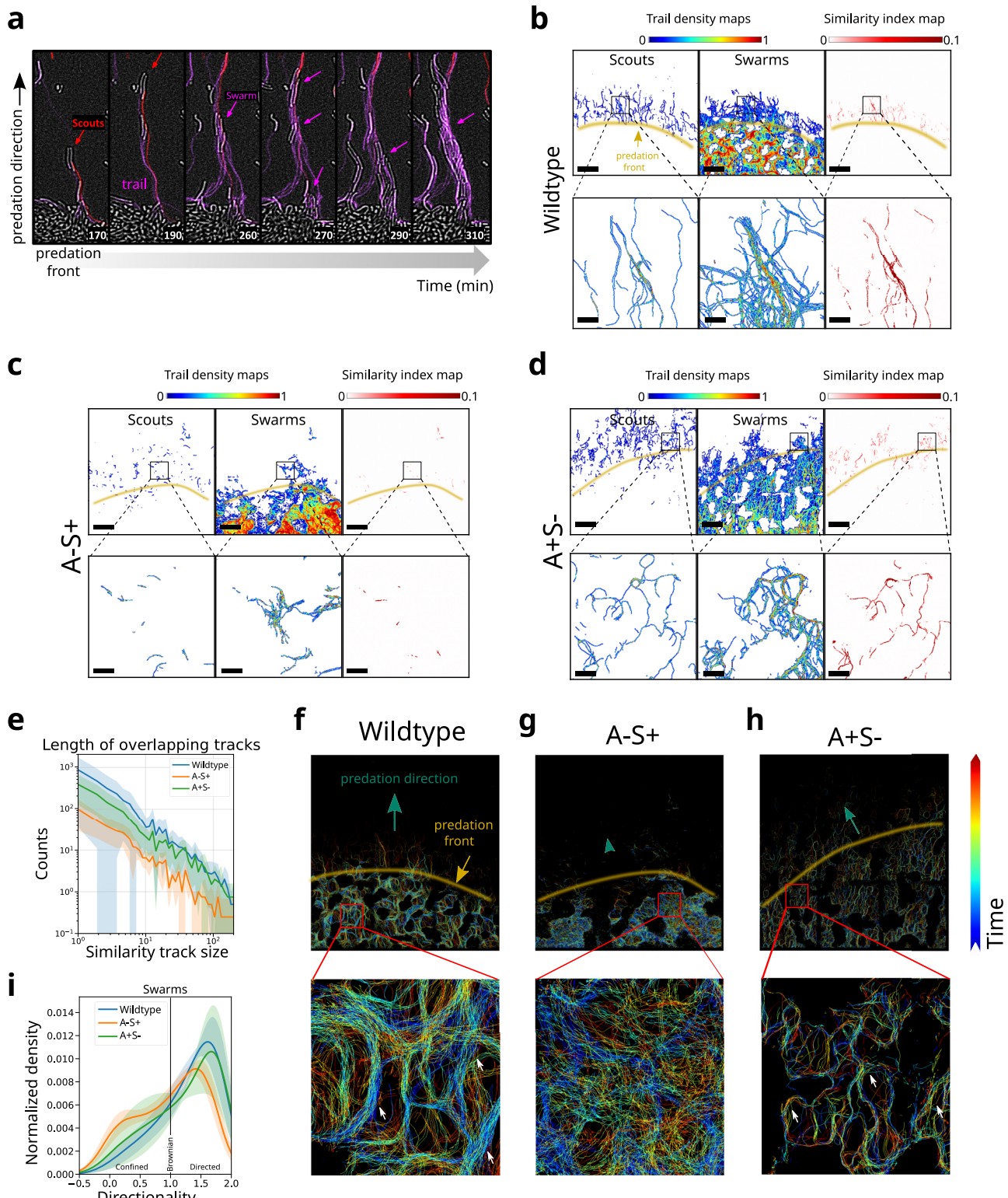

**Fig. 4 | A-motile cells drive collective cell movement along trails. a** Example of scout cells (red line) traveling away from an *E. coli* island (bright field) followed by swarms (purple lines). **b–d** Trail maps of scout (left) and swarm (middle) cells and similarity index map (right) for wild type (**b**), A-S+ (**c**), and A+S- cells (**d**). Yellow lines delimitate the predation front. Scalebars = 100 μm. Scalebars of zoomed areas = 20 μm. **e** Histogram of the length of overlapping tracks (Similarity track length) for wild type, A-S+ and A+S-. Shaded areas highlight the standard deviations from the mean (solid line) of six experimental replicates for the wild-type and four replicates for each mutant strains (see Fig. S4d for single replicate examples). **f–h** Overlays of all trajectories for wild type (**f**), S-motile (A-S+) cells (**g**), and A-motile (A+S-) cells (**h**) color-coded by time. Green arrows indicate the predation direction, yellow lines delimitate the predation front. White arrows in the zoom of the boxed areas point to examples of trails. **i** Histogram of movement directionality of swarm cell classes for wild type, A-motile (A+S-) and S-motile (A-S+) communities with directionality lower than 1 being confined motion, equal to 1 being Brownian motion and higher than 1 being directed motion. Shaded areas highlight the standard deviations from the mean (solid line) of six experimental replicates for the wild type and four replicates for each mutant strains (see Fig. S4h for single replicate examples).

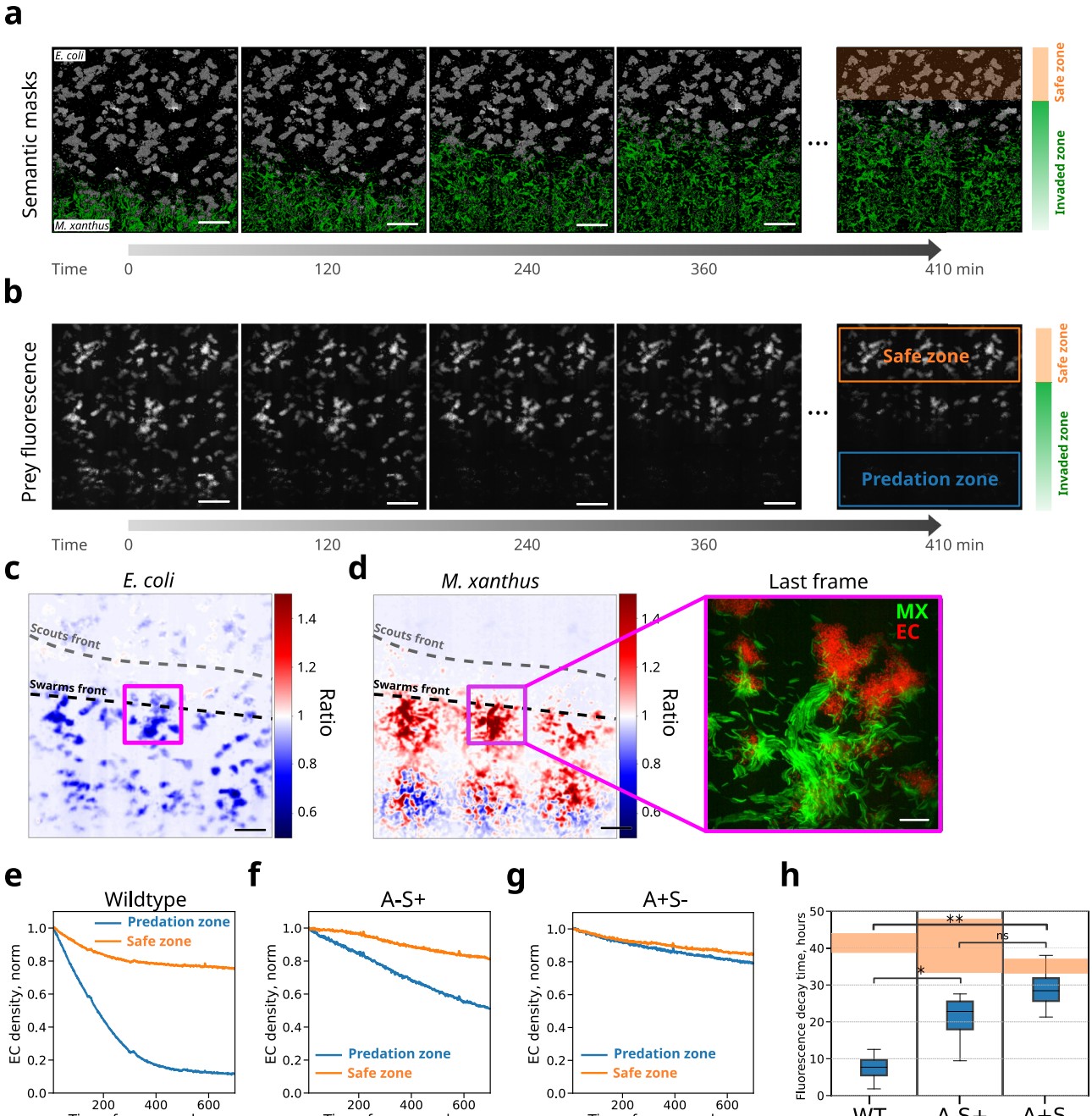

**Fig. 5 | A- or S-motility mutations affect predation efficiency. a** Predation over time at the forefront visualized with semantically segmented images containing the masks for *M. xanthus* (green) and *E. coli* (white). Two zones were defined: (i) where no invasion occurred (orange area) and (ii) where *M. xanthus* cells invaded *E. coli* (Invaded zone). Scalebars = 100 μm. **b** Total raw fluorescence signal from *E. coli* cells over time. Orange and blue boxed areas highlight the safe zone, where no predation occurred and the predation zone, where active predation is occurring, respectively. Scalebars = 100 μm. **c** Ratio of the last to the first image of the *E. coli* HU-mCherry raw fluorescence. Black and gray dashed lines represent a visual guide to the front of swarm and scout cell populations, respectively. Colorbar: blue represents regions where *E. coli* cells were killed. **d** Same as c but for *M. xanthus* OM-sfGFP. The right panel represents an overlay of the raw HU-mCherry images from *E.*

*coli* and OM-sfGFP from *M. xanthus* corresponding to the ROI represented by a purple square in the left image and in panel c. Colorbar: red represents regions with an enrichment in *M. xanthus* cells in the last time point, whereas blue regions represent depletion of *M. xanthus* cells. **e**–**g** Quantification of the total fluorescence signal from *E. coli* cells over time in the safe and predation zones for wild type (**e**), S-motile (A-S+) (**f**), and A-motile (A+S-) (**g**) predators. **h** Box plot summary of total fluorescence decay time from *E. coli* cells in the safe and predation zones (orange and blue shaded boxes, respectively) from six experimental replicates for the wild-type and four replicates for each mutant strains. Boxes show the median and the interquartile range, while whiskers represent the minimum and maximum values in the datasets. The statistics were calculated using a two-sided t-test. *p*-values were as follows: A-S+ vs A+S- = $1.10^{-1}$. A-S+ vs wild type = $3 \times 10^{-2}$. A+S- vs wild type = $3 \times 10^{-3}$.

movement of collective groups. Our data rather show that A-motility reinforces the exploratory behavior of both individual and collective groups while S-motility reinforces the cohesion of collective groups which may prompt whether the two motility engines are exclusively or concurrently activated in each individual cell.

Remarkably, individual cells can frequently merge into, or split from collective cell groups, highlighting the plasticity and highly dynamic behavior of *M. xanthus* cells during predation. This finding suggests that collective cell groups are not pre-assembled and that there is likely a low energetic barrier for group fusion or splitting. Thus,

groups can be made or unmade depending on the local environmental conditions to provide flexibility and adaptation. Interestingly, the transition frequencies between collective cell groups were in all cases symmetric, ensuring the long-term equilibrium of the system. This equilibrium may be perturbed to respond to local changes in prey distribution, providing another avenue for adaptation. Importantly, the removal of either motility apparatus did not change the ability of collective groups to fuse or split, but rather the transition frequencies and their final proportions. Thus, fine-tuning transition frequencies would provide a means to rapidly change the relative proportions of collective cell groups to adapt to the local ecosystem and its spatial organization. We hypothesize that this fine-tuning may involve modulation of reversal frequencies.

In non-predating conditions, A-motile cells secrete a slime trail that can become a preferred path for other cells[23–26]. Our data show that during active predation, trajectories of scout and swarm cells frequently overlap in space, suggesting that cues left by scouts are very local. Interestingly, these trajectories are revisited several times over long time periods (>hours), suggesting that these cues persist over time or are reinforced by successive passages. Thus, A-motility seems necessary for defining and encoding spatial memory cues on how to best explore the foraging space on the predation front. This consolidation and reuse of existing trails (guided by extracellular matrix components) could increase foraging efficiency and prey consumption, a process termed stigmergy[26,28]. In addition, our data show that wild-type swarms contain A-motile cells, and that swarms of A-motility mutants display perturbed speeds and directionality. Thus, A-motile cells within swarms potentially contribute to the directionality of swarms to provide the ability to navigate without a centralized system[29] which is known to increase the robustness and efficiency of reaching a target in biological[30] and in autonomous driving systems[29]. Future research, however, will be required to elucidate if and how A-motile cells may guide collective groups[31].

We are aware that the spatial organization of the prey community is likely to modulate the relative roles of A- and S-motilities during predation. For instance, A-motility may be more determinant than S-motility in close-knit prey communities, as the ability to penetrate the outer wall of these communities depends acutely on A-motility and contact-dependent killing[17]. Our data further reveal that after the first step of prey colony invasion by the scouts, the bulk of predation is performed by large groups of cells. Thus, the presence of cells with both motility systems in all collective cell groups enhances the ability of *M. xanthus* to adapt to the varying spatial organization and diversity of prey encountered in natural ecosystems.

Earlier studies established that A- and S-motilities enable *M. xanthus* to move on a wider range of substrates, suggesting that both machines may be needed to provide flexibility and adaptation to the physical properties of the local environment[27]. This arguably minor fitness gain may appear as insufficient to outweigh the evolutionary burden of simultaneously maintaining both motility machines. Our results, however, show that A- and S- motile cells work in unison to improve the efficiency of prey exploration and invasion and to lead to more efficient predation. We envision that the collective movement of A- and S-motile cells may enable *M. xanthus* communities to adapt their strategy to the defense and attack mechanisms of competing communities. All in all, these fitness advantages may largely outweigh the evolutionary costs associated with maintaining both motility systems.

## Methods

### Predation assays

Bacterial predation was established in laboratory conditions by setting up a predation assay as described in Rombouts *et al.*[19]. *E. coli* and *M. xanthus* cells were harvested from LB and CYE media, respectively. Next, cells were concentrated for 5 min at room temperature and resuspended in CF medium (10 mM MOPS (pH 7.6), 1 mM $KH_2PO_4$, 8 mM $MgSO_4$, 0.02% $(NH_4)_2SO_4$, 0.2% sodium citrate, and 0.015% bacto casitone peptone). *M. xanthus* cells were concentrated to an $OD_{600}$ of 5, *E.coli* cells to an $OD_{600}$ of 0.005. 1 µl cell suspensions were spotted at a distance of ~1 mm on CF 1.5% agar pads made with ultrapure agar (UltraPure Agarose 1000, Invitrogen). Predation assays were then placed onto a layer of CF 1.5% agar in a petri dish closed with parafilm to avoid agar pad evaporation and drying. Samples were incubated from 24 h to 48 h at 32 °C to allow *M. xanthus* cells to invade the *E. coli* colony. The list of strains used in this study can be found in Supplementary Table 1.

### Microscopy/fast time-lapse imaging

Fast time-lapse and hubble imaging was performed as described in Rombouts et al.[19]. In short, the predation assay sample was covered with an imaging coverslip and placed on a homemade fully-automated hardware-accelerated wide-field epifluorescence microscope built on a RAMM modular microscope system. A region of interest spanning an area of ~0.36 mm² was imaged through a ×60, 1.2NA objective by constructing a mosaic patchwork of 3 by 3 fields of view (FOVs) each of 2048 × 2048 pixels, overlapping by 200 pixels (1 pixel corresponding to 106 nm at the sample plane). Following a snake-like pattern, 3D stacks in the brightfield and fluorescence channels were acquired for each FOV (Exposure time of 50 ms) of the mosaic. By repeating this snake-like acquisition, an hours-long time-lapse series of the mosaic area could be captured. Generally, 700 time points were collected, each interspaced with 35–40 s. Robust acquisition of thousands of three-dimensional multicolor microscopy images was automated using Qudi-HiM, a modular software package written in Python[32].

### Treatment of fast time lapse data

As described in Rombouts et al.[19], the DCIMG image files were converted and sorted into tiff files with software from Hamamatsu. Tiff images of the fluorescent channel of *E.coli* were deconvolved with Huygens Professional version 20.04 (Scientific Volume Imaging, the Netherlands, https://svi.nl/). Deconvolved *E. coli* stacks were z-projected by calculating the standard deviation. 3D brightfield stacks were converted to 2D images by dividing each stack in 16 ROIs of 512×512 pixels, selecting automatically or manually the in-focus plane for each ROI and restitching the 16 ROIs (im_straighter.m or im_straighter_manual.m). 2D brightfield and *E. coli* fluorescence images were used as input for an in-house developed MATLAB code using a convolutional neural network with U-Net architecture for semantic segmentation (segment.m).

Segmented images were then used to reconstruct the mosaic image by tiling the 9 FOVs. The precise overlap between FOVs for tilling was calculated by image-based pixel-resolution cross-correlation (tile_calculation.py). Drift in time was corrected by aligning the mosaic images based on cross-correlation calculated from segmented images of stationary *E. coli* microcolonies (mosaic_drift_correction.py). *M. xanthus* segments were binarized and post-processing of E. coli masks ensured that there was no spatial overlap between *E. coli* and *M. xanthus* masks (builds_mosaic.m).

### Single-cell tracking

Single-cell trajectories were reconstructed with an in-house developed MATLAB pipeline (tracking.m). In short, pairwise tracks between *M. xanthus* cell masks in consecutive time points were reconstructed based on several mask parameters, such as cell area, cell length, and mask overlap area. The Analytical Hierarchy Approach was used to perform this task in densely populated regions where multiple masks could be linked. At last, pairwise tracks over all time points were linked together to reconstruct the full-length single-cell trajectories.

## Mean squared displacement calculation

To characterize directionality of bacterial movement, individual bacterial trajectories were analyzed with the Python *trackpy* package. For each track, the Mean Squared Displacement (MSD) was computed and the five first-time points were fitted with a power law. The resulting scaling exponent, alpha, was used to characterize the directionality of bacterial movement (from confined with alpha<1, brownian with alpha=1, to directed with alpha>1). Negative directionality indicates a decreasing displacement-time relationship, which can occur due to hindrance or confinement of cell movement. However, negative exponents should be interpreted cautiously as they can result from fitting limitations, artifacts, or tracking issues, and may not have significant biological meaning. The code used to calculate the MSD (002_Myxo_trackpy.py) can be found in the underline{bacto_tracker} repository.

## Calculation of gyration radii

Gyration radii were calculated by computing the center of mass of each trajectory and then quantifying the average distance of each point in the trajectory from the center of mass.

## Calculation of cell speeds

Instantaneous cell speeds were calculated using the straight distance traveled in the five frames before and five frames after a given time point, normalized by the time between 10 frames. The code used to calculate speed (004_matfiles_to_dataframe.py) can be found in the underline{bacto_tracker} repository.

## Voronoi tessellation

To measure the local density of *M. xanthus* cells, a Voronoi tessellation was performed with the Voronoi function of MATLAB. The Voronoi tessellation was calculated on the centers of the backbones of all masks that were included for tracking. Centers of gravity of the masks that were filtered for tortuosity and mask fusion based on branch points were included as well. For masks that contained a branchpoint in their backbone, the branchpoint was deleted, essentially breaking up the backbone. The centers of the newly generated backbones were calculated and included for the tessellation. For masks filtered out for tortuosity, the backbones' centers were also calculated and included for the tessellation. The inverse of the area of the polygon to which the mask belongs was used as a measure for local density, with large polygon areas for low cell density and small polygon areas for high cell density. The code used to calculate the local density of *M. xanthus* cells (function_Voronoi_backbone_OverlapFlag.m) can be found in the underline{bacto_tracker} repository.

## Long-range clustering

Long-range clustering of cells was performed by dilating the binary cell masks of *Myxococcus xanthus* cells using a $10 \times 10$ pixels kernel ($-1 \times 1 \mu m$). Then, merged cell masks were identified as clusters using the regionprops module of the skimage package in Python. For each identified cluster, its size was measured as the area covered by the non-dilated cell masks comprising each cluster and the number of cells per cluster was determined by dividing the total area of the cluster by the average area of a single cell. This approach was chosen to mitigate errors in single-cell segmentation, particularly in cases involving large groups of cells densely packed together. However, due to the heterogeneity in cell size, non-integer values occasionally arise for the number of cells per cluster, resulting in the absence of partial discretization along the x-axis of the histograms of Figs. 1f, j, k and 3d, e. The code used to perform long-range clustering (003_multiscale_segmentation.py) can be found in the underline{bacto_tracker} repository.

## Classes

Bacterial populations were categorized into three groups: loners, scouts, and swarms. Two criteria were used to determine the group to which a cell belongs to at each time point in the time-lapse: the Voronoi cell density, $V$, and the number of cells per cluster, $N$. For loners: $\log_{10}(V) \leq 4.5$ and $N \leq 2$; for scouts: $\log_{10}(V) \geq 4.5$ and $N \leq 20$; and for swarms: $\log_{10}(V) \leq 4.5$ and $2 < N$. For histograms of motile cells only, an additional criterion was used to select cells with an end-to-end track displacement higher than 2 pixels.

## Two-dimensional histograms

Two-dimensional data points histograms were computed using the Python Datashader package, with canvas mapping data to pixels as points. Histograms in Fig. 1 correspond to the mean of normalized histograms for each experimental condition (single experiments histograms are shown in Fig. S1). For histogram differences in Fig. 3, the normalized mean histograms of each condition were subtracted by taking the difference between the corresponding bins of the two histograms. The code used to reproduce the histograms in Figs. 1 and 3 (Figure_1FHI_3DE.py) can be found in the underline{bacto_tracker} repository.

## UMAP projection

Uniform Manifold Approximation and Projection (UMAP) was employed for dimensionality reduction and visualization[33]. A normalization step was performed on the input local density and cluster size features using the StandardScaler from scikit-learn. The UMAP algorithm was executed using a Number of Neighbors of 100 and a Minimum Distance of 0. UMAP's default values were used for other parameters, such as the distance metric (Euclidean) and the number of dimensions in the projected space (2D visualization). The UMAP projection results were visualized using Datashader. The embedding is provided at: https://osf.io/kc3sw/

## Transition analysis

The transitions between classes were calculated as follows. For each time point in the trajectories, the class of the cell was previously calculated as scout/loner/swarm. To avoid spurious transitions, a rolling filter over a 10-frame window was applied on each trajectory to keep only the most frequent class in the window. For each time point of the trajectories, the coordinates of the cell are replaced by that of the corners of a triangle corresponding to one of the three classes, with Gaussian noise added to reduce the overlap of the tracks. A two-dimensional histogram of the resulting trajectories is then computed using the Python package Datashader by summing the set of trajectories for each experimental condition. The code used to compute transitions between classes (Figure_2CDE.py) can be found in the underline{bacto_tracker} repository.

## AglZ foci detection

AglZ foci were automatically detected to highlight the position of AglZ complexes in single cells. For this, the raw fluorescent z-stacks were first band pass filtered to remove noise and low spatial frequencies in single planes, and then a local normalization was applied to equalize signal strength heterogeneities due to the Gaussian excitation profile. Finally, the four central images of the z-stack were summed and used to localize AglZ complexes as diffraction-limited spots using the DAOStarFinder utility from the Astropy package (https://www.astropy.org/).

## Similarity index map

To quantify the similarity between the trajectories of the scouts and the rest of the population, the trajectories of the two populations were split apart to map them on separated 2D arrays. Each array map was then binarized and used to compute a structural similarity index map with the Python Scikit-image package (sliding window of three pixels). Finally, the resulting similarity index map was binarized to extract the area of each portion of trajectory shared by the two populations. These areas of shared trajectories were used to quantify the amount of scout

trajectories shared with the rest of the bacterial population. The code used to calculate similarity index maps (Figure_4BCD.py) can be found in the bacto_tracker repository.

### Trail maps
Trail maps were computed as two-dimensional histograms using the Python Datashader package with line canvas mapping data to pixels. Histograms were computed by projecting all trajectories onto an 5880×5880 grid, aggregating the field of time for each trajectory coordinate by mean. The code used to calculate trail maps (Figure_1GJ_4BCDFGH.py) can be found in the bacto_tracker repository.

### Prey consumption
To quantify the consumption of prey cells by *M. xanthus* during invasion, the fluorescence of E. coli HU-mCherry was used. For each time point in the movies, the fluorescence intensity of the central plane of each z-stack was first normalized by the Gaussian profile of the excitation laser and then projected along the perpendicular direction of invasion. The mean intensity was then truncated into three equal parts to quantify E. coli HU-mCherry intensity changes in the portion of the field of view (FOV) that gets invaded by *M. xanthus* cells during the acquisitions (bottom part of the stitched FOV) and in a portion that does not get invaded during the acquisition (Top part of the stitched FOV), the central portion not being used. For the bottom and top parts of the FOV, the mean fluorescence intensity of E. coli HU-mCherry was quantified for each frame of the movies to characterize the disappearance of E. coli cells over time. The code used to calculate prey consumption (Figure_5EFG_preprocess.py and Figure_5EFG.py) can be found in the bacto_tracker repository.

### Reporting summary
Further information on research design is available in the Nature Portfolio Reporting Summary linked to this article.

## Data availability
Data generated in this study were deposited at our Open Science Framework project bacto_tracker with https://doi.org/10.17605/OSF.IO/UX4H9.

## Code availability
The code used for pre-processing, for semantic segmentation, and for building tracks is accessible at https://github.com/NollmannLab/bacto_tracker. Networks, example images, and an archived version of bacto_tracker can be found at https://osf.io/ux4h9/ under https://doi.org/10.17605/OSF.IO/UX4H9 (components: *UNET networks, Data*). The code used for post-processing and for building the figures in this manuscript is accessible at Rombouts_et_al (permanent link: https://doi.org/10.17605/OSF.IO/UX4H9). Data was acquired using qudi-HiM[32]. The current version of qudi-HiM is found at https://github.com/NollmannLab/qudi-HiM, and an archived version at https://zenodo.org/record/6379944 (https://doi.org/10.5281/zenodo.6379944). Qudi-HiM has been added to the RRID database (record ID: SCR_022114).

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

## Acknowledgements

This project was funded by the European Union's Horizon 2020 Research and Innovation Program (Grant ID 724429, M.N; Grant ID 885145, T.M.) and under the Marie Skłodowska-Curie Grant Agreement No 721874 (SPM2.0). We acknowledge the Bettencourt-Schueller Foundation for their prize 'Coup d'élan pour la recherche Française', the France-BioImaging infrastructure supported by the French National Research Agency (grant ID ANR-10-INBS-04, "Investments for the Future").

## Author contributions

S.R., T.M., and M.N. conceived the study and the design. S.R. and A.M. acquired the data. S.R., J-B.F., A.L, and A.M. analyzed the data. J-B.F. built the microscope. S.R., J-B.F., and A.L. wrote the software. S.R., A.M., A.L., J-B.F., T.M., and M.N. interpreted the data. M.N. and S.R. wrote the first manuscript draft. M.N., S.R., T.M., A.M., and A.L. participated in manuscript editing. T.M. and M.N. supervised the study and acquired funds.

## Competing interests

The authors declare no competing interests.
