## [Peer Review File · Nature Communications]

Multi-scale dynamic imaging reveals that cooperative motility behaviors promote efficient predation in bacteriaREVIEWER COMMENTS

Reviewer #1 (Remarks to the Author):

In the manuscript entitled “Multi-scale dynamic imaging reveals that cooperative motility behaviors promote efficient predation in bacteria”, the authors reported a study on the motilities of *Myxococcus xanthus* during predation. With the advanced imaging techniques, the authors show different roles of A-motility and S-motility during the predation process. More interestingly, A-motile cells are present in all cell classes, in contrary to the common picture that the A-motility is exclusively to scouts. The authors also show that efficient predation requires both A-motility and S-motility. Overall, I think this work is very interesting and the analysis is fantastic. My main concern is that the study shows interesting results which are all based on phenomenological observations, but is lack of mechanisms to explain what the authors observed. For example, in this work, cells are classified into three types (scouts, swarm, loners) based on cells’ local density and associated cluster size. How do these three types correlate to the physiological states of a cell? When cells transit among these different classes, do their physiological state also change? What signal/substance corresponds to the transition between different classes?

Other comments/suggestions :

- 1) The three types are one of key concepts in this work. The authors may consider to show some examples at the single-cell level for each different type of classes.
- 2) It might be better to have a schematic drawing to show the set-up and how the samples were imaged. There are not enough details in Methods to repeat such measurements.
- 3) The authors stated that “We observed that scouts in A-S+ colonies traveled considerably shorter distances than wild-type (Figs. 1G,J)”. The authors may consider to make a better comparison by showing the radius of gyration quantitatively for each case.
- 4) In Figure 2A, what is the duration of each state that a cell can stay before transition?
- 5) For the clustering, the authors used a 10*10pixels kernel. Why did the author choose this value? How do the results depend on the chosen value?
- 6) In figures 1L, 4I, what is the meaning of the negative directionality?

Reviewer #2 (Remarks to the Author):

In this paper Rombouts et al. investigate the roles of the adventurous (A, gliding) and the social (S, twitching) motilities during predation of *E. coli* colonies in the myxobacterium *Myxococcus xanthus*.

Using a large scale imaging setup, they follow the invasion dynamics by the *M. xanthus* colony at single cell resolution, for the wild-type, A-motile-only and S-motile-only genotypes. They found, in general agreement with previous works, that cells can be classified in 3 types/roles: fairly isolated scouts, alone or in small groups, at the forefront of the colony, densely packed swarm cells grouped in large clusters further at the back and lone cells in fairly dense regions but that detached from the clusters. Individual cells switch roles fairly frequently throughout the predation. Contrary to previous observations on non-predating *M. xanthus* swarms, they found that A-motile and S-motile cells were both able to assume all roles, and that they intermixed while doing so. They however have different effects on the population behavior. The A-motile cells switched roles more frequently, produced more lone cells and scouts, displayed more directional motion but produced less cohesive swarms. A-motile scouts also appeared to leave trails that swarms were then able to follow when advancing in the colony. Finally, S-motile cells appear to be necessary to enhance the efficiency of prey lysis, and thus consumption at the back of the swarm. The work clarifies and nuances the roles of the two motility types, which have been maintained together throughout evolution in *M. xanthus*, during a presumably ecologically important part of its life. Although I do have a few comments and questions that must be addressed by the authors, I think it is already a fairly sound work. The conclusions are already decently well supported by the data. It brings interesting new information to the field. So I think it should eventually be publishable in Nature communications.

Main points:

1 I find the classification in scout/loner/swarm to have a fairly arbitrary definition. When looking at the histograms of Fig 1F,H,I and Fig S1 one gets the sense that there is indeed an underlying multi-lobe distribution, but which does not necessarily overlap with the classification of the authors. I wonder if the authors tried defining categories of cells in a more organic manner, with a PCA-based (or similar) computer classification for example. I think this should at least be discussed more. In particular, the robustness of the results to the exact choice of boundaries should a minima be determined. I also wonder what would happen if other parameters like the position relative to the swarm front would be taken into account.

2 a major issue with the study is the complete lack of quantification of the statistical significance of the various differences between strains (except Fig 5H), although the variability between replicates, and therefore the error bars, seems fairly large. This should be carried out for all quantitative comparisons in all figures.

3 Figure 2: it would be nice to quantify the fraction of cells in each state for each mutant, and whether the possible differences are significant. This would strengthen the point that A-motile cells are more “exploratory”

4 Figure 4, Slime trail: It is very likely that the slime trail is responsible for the guiding effect of A-cells. Is this behavior lost in a non-secreting (or reduced secretion) mutant, or with appropriate surface treatment? I feel this is fairly easy to address and would strengthen Fig. 4 results.

5 Fig 5: it seems that most of the killing is done where the swarm is located. However, it could also be possible that the time it takes for a cell to lyse masks the effectivity of the scouts in killing. Was this time measured and how does it compare with the speed of the swarm front?

Secondary points:

6 It is strange to me that the Voronoi area-cluster size histograms are not partially discretized along the x-axis. This suggests that there are non-integer numbers of cells per cluster. Please clarify how the histogram is computed to explain this.

7 Fig 1I: what is the red arrow?

8 Fig 2B / Fig S2 : it would be good to plot the 3 histograms on the same graph for easier comparison.

9 Fig 4E and Fig S4G: Plotting the distributions in log scale along the y-axis would make the data more readable.

10 Figure S5B & D the frame seems to be the same at all time points.

11 experimental methods: please provide the magnification and NA of the objective, as well as the exact pixel size of the camera.

12 methods of classification: the definition of V is ambiguous, I assume it is the Voronoi area and not the cell density (ie its inverse). The definition of the scouts seems to have a typo.

13 The "Figures" codes are not available on Github yet

14 several references, eg Refs 1, 7, 15, 17 are incomplete

Reviewer #1 (Remarks to the Author):

In the manuscript entitled “Multi-scale dynamic imaging reveals that cooperative motility behaviors promote efficient predation in bacteria”, the authors reported a study on the motilities of Myxococcus xanthus during predation. With the advanced imaging techniques, the authors show different roles of A-motility and S-motility during the predation process. More interestingly, A-motile cells are present in all cell classes, in contrary to the common picture that the A-motility is exclusively to scouts. The authors also show that efficient predation requires both A-motility and S-motility. Overall, I think this work is very interesting and the analysis is fantastic.

We thank the referee for their careful examination of the manuscript. We have addressed these points by implementing changes to the manuscript and figures, which are outlined below.

Main points:

1.0 *My main concern is that the study shows interesting results which are all based on phenomenological observations, but is lack of mechanisms to explain what the authors observed. For example, in this work, cells are classified into three types (scouts, swarm, loners) based on cells' local density and associated cluster size. How do these three types correlate to the physiological states of a cell? When cells transit among these different classes, do their physiological state also change? What signal/substance corresponds to the transition between different classes?*

We would like to thank the reviewer for their insightful comments on our manuscript.

In our study, we have used a novel high-throughput imaging method to monitor single-cell behavior in a biofilm during bacterial predation. The ability of our method to follow the spatio-temporal dynamics of single cells was critical to discover a frequent exchange of single cells within collective groups and to quantitatively characterize it, which was hitherto unknown.

To shed light on the mechanism, we performed experiments in motility mutants, which were key to concluding that both kinds of motile cells can exchange between groups, with transition rates that were motility-system-dependent and that determined the number of cells per group at equilibrium.

To further understand the mechanisms involved, we calculated the speeds and directionality of each group for wildtype and mutants. This suggested that scout groups direct the movement of swarms. This was further consolidated by analyzing how different cell groups followed each other in wildtype and in motility mutants. Overall, we convincingly showed that scouts play a role in exploring and guiding swarms during predation.

Finally, to shed light on the mechanism of prey killing, we quantified how *E. coli* was preyed on by different collective groups and how mutants using different motilities performed at this process. This allowed us to demonstrate that A- and S-motile cells act in synergy to increase predation efficiency.

All in all, we think these experiments and analyses not only unveiled unexpected observations but also provided considerable mechanistic insight.

We agree that further experiments will be required to understand whether the collective groups display different physiological states, and if there is any internal signal or substance that may trigger transitions between classes. However, we note that answering these additional questions would require years of development and research. For instance, we would need to develop sophisticated methods to follow the dynamics of single cells and then restore the physiological state of the tracked cells with spatial information. We note that spatial transcriptomics in *Myxococcus* biofilms is not reported in the literature. Even for eukaryotic systems, where these spatial transcriptomics have been used for years, the coupling of dynamic and spatial omics information is not yet possible. Similarly, challenging new technology developments and validations would be needed to detect and identify substances that may be exchanged between dynamically single cells in dense environments. Because of these reasons, we think it is reasonable to investigate these mechanisms in follow-up studies.

In spite of this, we made a considerable effort to contribute answers to the reviewer. For this, we measured proxies of physiological states from single-cell images, such as cell length, perimeter, and eccentricity. The distribution of cell lengths and widths was frequently used in the literature to estimate whether cells are growing and dividing (Belliveau et al, 2021, PMID 34214468). We performed these measurements for each class, but we did not observe significant differences (Figure PBP1, below). These analyses are consistent with cells in scout, loner, or swarm groups displaying similar rates of growth. In addition, prompted by questions from other reviewers, we quantified the fraction of cells in each state for each mutant condition (see 2.3 below). This analysis strengthens the conclusion that A-motile cells are more exploratory than S-motile cells.

Figure PBP1: From left to right: normalized probability density histograms of cell length, perimeter, and eccentricity for scouts, loners, and swarms (blue, orange, and green histograms respectively) on cumulated experiments in wild-type conditions. One pixel represents 105 nm.

Belliveau NM, Chure G, Hueschen CL, Garcia HG, Kondev J, Fisher DS, Theriot JA, Phillips R. Fundamental limits on the rate of bacterial growth and their influence on proteomic composition. *Cell Syst.* 2021 Sep 22;12(9):924-944.e2. doi: 10.1016/j.cels.2021.06.002. Epub 2021 Jul 1. PMID: 34214468; PMCID: PMC8460600.

Secondary points:

1.1 The three types are one of key concepts in this work. The authors may consider to show some examples at the single-cell level for each different type of classes.

We thank the reviewer for this suggestion and we have generated examples of single cells color-coded based on classes (new Fig.1 panel 2h).

New Fig.1 panel 2h: Example of single cells masks color-coded with cell classes in wild-type conditions. *Escherichia coli* cell masks are blue, *M. xanthus* scouts, loners and swarms are colored in yellow, green and red, respectively. Scale bar is 10 μ m.

In addition, we modified the text as follows:

“The three classes are as follows: (1) scouts are small groups of cells (1-20) isolated from the main colony, typically localizing ahead of the forefront of the invading wave; (2) swarm cells lie within cell clusters and are always closely packed with other cells; and (3) loners cells lie close to other cells within the colony front (see snapshots and trajectories in Figs. 1h-i).”

1.2 *It might be better to have a schematic drawing to show the set-up and how the samples were imaged. There are not enough details in Methods to repeat such measurements.*

We appreciate the reviewer's suggestion regarding the inclusion of a schematic drawing to illustrate the experimental setup and imaging process. However, due to the complexity of the method, it was not feasible to provide sufficient details within the main article. To address this limitation, we have appropriately cited a separate paper by Rombouts et al (ref 19) in our methods section that provides a comprehensive description of the experimental setup and imaging procedures. This additional paper contains the necessary information to replicate the measurements and serves as a detailed reference for interested readers.

In addition to the method paper, we also better referenced Qudi-HiM, our published and open-source acquisition software package that automates acquisition of our time-lapse data.

The method section now reads:

“Microscopy/Fast time-lapse imaging

Fast time-lapse and hubble imaging was performed as described in Rombouts *et al.* ¹⁹. In short, the predation assay sample was covered with an imaging coverslip and placed on a homemade fully-automated hardware-accelerated wide-field epifluorescence microscope built on a RAMM modular microscope system. A region of interest spanning an area of approximately 0.36 mm² was imaged through a 60X, 1.2NA objective by constructing a mosaic patchwork of 3 by 3 fields of view (FOVs) each of 2048x2048 pixels, overlapping by 200 pixels (1 pixel corresponding to 106 nm at the sample plane). Following a snake-like pattern, 3D stacks in the brightfield and fluorescence channels were acquired for each FOV (Exposure time of 50 ms) of the mosaic. By repeating this snake-like acquisition, an hours-long time-lapse series of the mosaic area could be captured. Generally, 700 time points were collected, each interspaced with 35-40 seconds. Robust acquisition of thousands of three-dimensional multicolor microscopy images was automated using Qudi-HiM, a modular software package written in Python ³².”

32. Barho, F. et al. Qudi-HiM: an open-source acquisition software package for highly multiplexed sequential and combinatorial optical imaging. *Open Research Europe* vol. 2 46 Preprint at <https://doi.org/10.12688/openreseurope.14641.2> (2022).

19. Rombouts, S., Fiche, J.-B., Mignot, T. & Nollmann, M. *bacto_tracker*: a method for single-cell tracking of *M. xanthus* in dense and multispecies colonies. *Open Res Europe* 2, 136 (2022).

1.3 *The authors stated that “We observed that scouts in A-S+ colonies traveled considerably shorter distances than wild-type (Figs. 1G,J)”. The authors may consider to make a better comparison by showing the radius of gyration quantitatively for each case.*

We thank the reviewer for their suggestion and have now quantified the gyration radius of scouts trajectories for wildtype and mutants (Fig. S1f below). These new analyses confirm that scouts in A-S+ colonies explore smaller areas (median gyration radii: 0.97 μm) as compared to wildtype (1.85 μm) or A+S- colonies (1.65 μm). We have now included this data in Fig. S1f.

Figure S1f: Histogram of gyration radii of scouts cells trajectories in wildtype, A-motile (A+S-) and S-motile (A-S+) communities. Shaded areas highlight the standard deviations from the mean (solid line) of six experimental replicates for the wild-type and four replicates for each mutant strains.

The main text and methods section now references this new analysis:

“We observed that scouts in A-S+ colonies traveled considerably shorter distances than wild-type (Figs. 11 and S1f), ...”

“Calculation of gyration radii

Gyration radii were calculated by computing the center of mass of each trajectory and then quantifying the average distance of each point in the trajectory from the center of mass.”

1.4 In Figure 2A, what is the duration of each state that a cell can stay before transition?

In the original manuscript, we refrained from presenting residence times because our method does not yield unbiased estimates for these measurements due to two main limitations: (1) on average, we can monitor cells for approximately 30 frames (Rombouts et al, 2022), which may be shorter than the actual time cells spend in each state. This means that measured residence times in each state may be underestimated. (2) our tracking efficiency is higher for isolated cells (e.g. scouts, loners) than for densely packed cells (e.g. swarms). Because of this, residence times may exhibit biases between cell classes. While characterization of residence times could have provided additional information, we do not think it is necessary to provide additional mechanistic insight.

1.5 For the clustering, the authors used a 10*10pixels kernel. Why did the author choose this value? How do the results depend on the chosen value?

The dilation kernel size of 10x10 pixels (~1x1 μm) for dilating single-cell masks during long-range clustering was carefully chosen to distinguish between cells that are fortuitously in proximity to one another from cells that are closely associated together in groups. By using a dilation kernel size approximately equal to the width of a single cell body, we aimed to achieve a balance between capturing meaningful cell clusters and avoiding the overestimation of clustered cells due to chance proximity.

To assess the robustness of the kernel size criterion for cell clustering, we conducted an analysis on a single frame, considering four different dilation kernel sizes: 5, 10, 15, and 20 pixels. We computed the statistics of the number of cells classified as scouts, loners, and swarms using these different kernel sizes (see Fig. S1b below). The analysis revealed that the number of scout cells remained unchanged across the different dilation kernel sizes, as these cells tend to be located far away from the main colony. On the other hand, the swarm class absorbed most of the neighboring loner cells as the kernel size increased. It is important to note that the statistics of the swarm class

remain relatively unaffected due to their initial large number of cells, while larger kernel sizes primarily affected the loner class, which was not the main object of our study.

In short, this analysis allowed us to verify the stability of the kernel size criterion for clustering cells and its impact on the classification of different cell categories. These results support the reliability and appropriateness of the chosen kernel size for our clustering approach.

Figure S1b: Effect of multiscale clustering dilation Kernel size parameter on the classification of cells. Histograms show the number of cells in either scout, loner or swarm classes for four distinct dilation kernel size values.

We have now included this new analysis in Figure S1b and referenced it in the main text:

The three classes are as follows: (1) scouts are small groups of cells (1-20) isolated from the main colony, typically localizing ahead of the forefront of the invading wave; (2) swarm cells lie within cell clusters and are always closely packed with other cells; and (3) loners cells lie close to other cells within the colony front (see snapshots and trajectories in Figs. 1h-i). We note that while scouts segregated into a well-defined cluster in the UMAP representation (Fig. 1g), loners and swarms displayed a partial overlap. Nonetheless, changes in the analysis and classification thresholds had moderate effects on the classification results (Figs. S1b-d).

1.6 In figures 1L, 4I, what is the meaning of the negative directionality?

The directionality values in this context represent the scaling exponent of the power law used to fit the Mean Squared Displacements (MSD) of individual cell trajectories. A negative directionality indicates a decreasing relationship between displacement and time. This occurs when the movement of cells is hindered or confined, possibly due to physical barriers or interactions with the environment.

However, it is essential to interpret negative exponent power law fits cautiously, as they can also arise from fitting limitations or artifacts. For instance, short trajectories, trajectories with reversing events, or trajectories with minimal movements can lead to inaccurate fits and result in negative exponents while cell segmentation or tracking limitations may result in artifactual cell position fluctuations. In such cases, the negative exponent values may not carry significant biological meaning and should be interpreted with caution.

This is now explained in the methods section:

Mean squared displacement calculation

To characterize directionality of bacterial movement, individual bacterial trajectories were analyzed with the Python *trackpy* package. For each track, the Mean Squared Displacement (MSD) was computed and the five first-time points were fitted with a power law. The resulting scaling exponent, alpha, was used to characterize the directionality of bacterial movement (from confined with $\alpha < 1$, Brownian with $\alpha = 1$, to directed with $\alpha > 1$). Negative directionality indicates a decreasing displacement-time relationship, which can occur due to hindrance or confinement of cell movement. However, negative exponents should be interpreted cautiously as they can result from fitting limitations, artifacts, or tracking issues, and may not have significant biological meaning. The code used to calculate the MSD (002_Myxo_trackpy.py) can be found in the bacto_tracker repository.

Reviewer #2 (Remarks to the Author):

In this paper Rombouts et al. investigate the roles of the adventurous (A, gliding) and the social (S, twitching) motilities during predation of E. coli colonies in the myxobacterium Myxococcus xanthus. Using a large scale imaging setup, they follow the invasion dynamics by the M. xanthus colony at single cell resolution, for the wild-type, A-motile-only and S-motile-only genotypes. They found, in general agreement with previous works, that cells can be classified in 3 types/roles: fairly isolated scouts, alone or in small groups, at the forefront of the colony, densely packed swarm cells grouped in large clusters further at the back and lone cells in fairly dense regions but that detached from the clusters. Individual cells switch roles fairly frequently throughout the predation. Contrary to previous observations on non-predating M. xanthus swarms, they found that A-motile and S-motile cells were both able to assume all roles, and that they intermixed while doing so. They however have different effects on the population behavior. The A-motile cells switched roles more frequently, produced more lone cells and scouts, displayed more directional motion but produced less cohesive swarms. A-motile scouts also appeared to leave trails that swarms were then able to follow when advancing in the colony. Finally, S-motile cells appear to be necessary to enhance the efficiency of prey lysis, and thus consumption at the back of the swarm. The work clarifies and nuances the roles of the two motility types, which have been maintained together throughout evolution in M. xanthus, during a presumably ecologically important part of its life. Although I do have a few comments and questions that must be addressed by the authors, I think it is already a fairly sound work. The conclusions are already decently well supported by the data. It brings interesting new information to the field. So I think it should eventually be publishable in Nature communications.

We thank the reviewer for their thorough review of the manuscript and their suggestions for improvement. We have taken the reviewer's suggestions into account and made appropriate revisions to the manuscript and figures, which are outlined below.

Main points:

2.1 *I find the classification in scout/loner/swarm to have a fairly arbitrary definition. When looking at the histograms of Fig 1F,H,I and Fig S1 one gets the sense that there is indeed an underlying multi-lobe distribution, but which does not necessarily overlap with the classification of the authors. I wonder if the authors tried defining categories of cells in a more organic manner, with a PCA-based (or similar) computer classification for example. I think this should at least be discussed more. In particular, the robustness of the results to the exact choice of boundaries should a minima be determined. I also wonder what would happen if other parameters like the position relative to the swarm front would be taken into account.*

We acknowledge the reviewer's concerns regarding the classification of cells into scout, loner, and swarm categories. We agree that the definitions we provided for these categories may appear somewhat arbitrary. Most previous studies primarily relied on the assumption that a particular collective behavior was associated with a specific motility engine: scouts with A-motility and swarms with S-motility. However, the literature lacks quantitative guidelines for defining these specific classes. In our original submission, we set criteria to define classes best matching the qualitative definition given in the literature: scouts are cells that detach from the colony to explore new territories, while swarms are groups of cells densely packed together. We defined loners as individual cells that did not fit in the other cell groups previously used in the literature.

We addressed this important comment in three complementary ways. First, we tested the robustness of this classification by exploring different parameter sets. In a previous analysis (see answer **1.5** above), we tested how kernel size affected the robustness of our classification, and found that the specific choice of kernel size leaves the swarm and scout classes relatively unaffected.

Next, we changed the thresholds for the number of cells per cluster and the local cell density, and analyzed whether these changes affected our classification. Decreasing the number of cells per scout group had minimal effects on their classification, whereas increasing this number primarily affected the statistics of swarm cells at the predation forefront. These results are summarized in Fig. S1c (below), where the original set of parameters is highlighted by a red box:

Fig. S1c. Spatial occupation of scout (top row) and swarm (bottom row) trajectories for wildtype conditions at the predation forefront for distinct threshold values of cells per cluster (N). Scalebars = 100 μm . The red boxes represent the conditions used for the analyses in this article.

Next, we analyzed the impact of different local density thresholds. These results are summarized in Fig. S1d (below) where the original set of parameters is highlighted by a red box. As expected, decreased local density thresholds ($\log_{10}(V) \geq 4$) led to an increase in the number of scouts, a corresponding decrease in loners, and no visible change for swarms. We note that the overall track characteristics and spatial distributions for the three classes remained similar (compare left column and middle column in Fig. S1d below). Similarly, an increase in the threshold ($\log_{10}(V) \geq 5$) resulted in a decrease in scout cells, a complementary increase in loner cells, and no visible change for the swarm population. Again, the track properties and spatial distributions remained poorly affected.

Fig. S1d: Spatial occupation of scout (top row), loner (middle row) and swarm (bottom row) trajectories for wildtype conditions at the predation forefront for distinct threshold values of Voronoi areas ($\log_{10}(V)$). Scalebars = 100 μm . The red boxes represent the conditions used for the analyses in this article.

The second approach to assess the robustness of our classification method was to examine whether changes in threshold values affected quantifications of cell speeds and directionalities for scouts (Figs. S1k-n below, original parameters highlighted in red). Remarkably, we observed that the distributions of speed and directionality for all classes were minimally affected by changes in the classification parameters. Thus, we conclude that small variations in the classification thresholds have limited effects on the dynamic behaviors of the three cell classes. These results underscore the robustness of our quantification approach and suggest that the classification criteria employed are reliable in capturing meaningful distinctions between the cell populations.

Fig. S1k-n: (S1k) Histogram of instantaneous speed of scout cells in wildtype conditions for distinct threshold values of cells per cluster (N). Shaded areas highlight the standard deviations from the mean (solid line) of six experimental replicates. The red text represents the parameter used for the analyses in this article. (S1l) Histogram of instantaneous speed of scout cells in wildtype conditions for distinct threshold values of Voronoi areas ($\log_{10}(V)$). Shaded areas highlight the standard deviations from the mean (solid line) of six experimental replicates. The red writings represent the conditions used for the analyses in this article. (S1m) Histograms of directionality of scout cells in wildtype conditions for distinct threshold values of cells per cluster (N). Shaded areas highlight the standard deviations from the mean (solid line) of six experimental replicates. The red writings represent the conditions used for the analyses in this article. (S1n) Histogram of directionality of scout cells in wildtype condition for distinct threshold values of Voronoi areas ($\log_{10}(V)$). Shaded areas highlight the standard deviations from the mean (solid line) of six experimental replicates. The red writings represent the conditions used for the analyses in this article.

We repeated this analysis for swarms, where we varied the thresholds for local density and cluster size and evaluated the impact in speeds and directionality (see Figs. S3c-d and Figs. S4j-k below, original parameters highlighted in red). As for scout cells, these distributions were robust to changes in the classification parameters.

Figs. S3c-d, S4j-k: Robustness of single cells classification parameters. (S3c) Histogram of instantaneous speed of swarms cells in wildtype conditions for distinct threshold values of cells per cluster (N). Shaded areas highlight the standard deviations from the mean (solid line) of six experimental replicates. The red writings represent the conditions used for the analyses in this article. (S3d) Histogram of instantaneous speed of swarms cells in wildtype conditions for distinct threshold values of Voronoi areas ($\log_{10}(V)$). Shaded areas highlight the standard deviations from the mean (solid line) of six experimental replicates. The red writings represent the conditions used for the analyses in this article. (S4j) Histograms of directionality of swarms cells in wildtype conditions for distinct threshold values of cells per cluster (N). Shaded areas highlight the standard deviations from the mean (solid line) of six experimental replicates. The red writings represent the conditions used for the analyses in this article. (S4k) Histogram of directionality of swarms cells in

wildtype condition for distinct threshold values of Voronoi areas ($\log_{10}(V)$). Shaded areas highlight the standard deviations from the mean (solid line) of six experimental replicates. The red writings represent the conditions used for the analyses in this article.

The third approach to determine the validity of our classification was to explore the use of dimensionality-reduction techniques, as suggested by the reviewer. For this, we used Uniform Manifold Approximation and Projection for Dimension Reduction (UMAP) to objectively test whether the different cell classes obtained by our classification approach segregated in a low-dimensional embedding (see Fig. 1g below).

First, we tested whether local density or cluster size parameters were sufficient to label cells that were segregated in the UMAP space (Fig. 1g, left and center panel). In fact, neither of these single parameters was enough as both displayed a continuous change within the UMAP space.

Next, we color-coded each single cell as either scout, loner or swarm according to our classification that made use of both local density and cluster size information. In this case, we observed that single scout cells clearly segregated from loners and swarms in the UMAP (Fig. 1g, right panel). The spatial segregation between loner and swarm cells was incomplete, but loner cells still occupied a distinct region of the map. The partial overlap between these two classes is not surprising, as small groups of swarm cells splitting from a swarm would have similar properties (local density, cluster size) as loners.

Fig. 1g: Dimensionality reduction analysis of cluster size and Voronoi area parameters with Umap. Left, middle and right panels represent aggregated single cells data points color-coded with cluster size, Voronoi area values, or classes as defined by our criteria, respectively.

We have now incorporated these complementary analyses into the manuscript providing a clearer explanation of the observed effects of the classification parameters on the cell populations and their associated quantifications:

“To verify the validity of this classification, we turned to Uniform Manifold Approximation and Projection for Dimension Reduction (UMAP). First, we tested whether local density or cluster size were sufficient to label cells that were segregated in the UMAP space (Fig. 1g, left and center panel). In fact, neither of these single parameters was enough as both displayed a continuous change within the UMAP space. Next, we color-coded each single cell as either scout, loner or swarm according to our classification that made use of both local density and cluster size information. In this case, we observed that single scout cells clearly segregated from loners and swarms in the UMAP (Fig. 1g, right panel). The spatial segregation between loner and swarm cells was incomplete, but loner cells still occupied a distinct region of the map. The partial overlap between these two classes is not surprising, as small groups of swarm cells splitting from a swarm would have similar properties (local density, cluster size) as loners. The three classes are: (1) scouts which represent small groups of cells (1-20) isolated from the main colony, typically localizing ahead of the forefront of the invading wave; (2) swarm cells which lie within cell clusters and are always closely packed with other cells; and (3) loners cells which lie close to other cells within the colony front (see snapshots and trajectories in Figs. 1h-i). We note that while scouts segregated into a well-defined cluster in the UMAP representation (Fig. 1g), loners and swarms displayed a partial overlap. Nonetheless, changes in the analysis and classification thresholds had moderate effects on the classification results (Figs. S1b-d).”

2.2 A major issue with the study is the complete lack of quantification of the statistical significance of the various differences between strains (except Fig 5H), although the variability between replicates, and therefore the error bars, seems fairly large. This should be carried out for all quantitative comparisons in all figures.

We apologize for the lack of quantification of the statistical significance between experimental conditions. We have now added supplementary materials with figures illustrating the same distributions as in the main figures but in the format of boxplots with statistical tests (Figs. S1i and S2b). These new quantifications show that changes in the speed distributions of scouts between wildtype and A-S+ are significant, while these distributions are statistically similar between wildtype and A+S- (Fig. S1i). In addition, we observe that the distribution of speeds for the swarm population significantly changes between wildtype and A-S+ and is less pronounced between wildtype and A+S-

Figure S1i and S2b: Box plots summaries of the speed of scout and swarm cells in wildtype (blue boxes), S-motile (A-S+, orange boxes) and A-motile (A+S-, green boxes) communities. Each boxplot represents the statistics of individual experiments. The statistics were calculated using a t-test.

Similarly, we constructed boxplots to compare the distributions of directionality between replicates and strains (Figs. S1j, S4i). We observed that the directionality of scouts and swarms are significantly affected in a A+S- strain with respect to wildtype, while these are not significantly unaffected in a A+S- strain.

Figure S1j, S4i: Box plot summaries of the directionality of scout (left panel) and swarm (right panel) cells in wildtype (blue boxes), S-motile (A-S+, orange boxes) and A-motile (A+S-, green boxes) communities. Each boxplot represents the statistics of individual experiments. The statistics were calculated using a t-test.

We have amended the text to incorporate these new analyses:

“Notably, A-S+ scouts displayed a marked reduction in speed (Figs. 1m, S1g), and a reduction in directed motion counterbalanced by a gain in Brownian and confined movements (Fig. 1n and Fig. S1h). This result is

in line with the finding that during colony expansion single, isolated *M. xanthus* cells (i.e. scouts/loners) move only if they carry a complete A-motility system⁸. We also note that wild-type scouts reach higher speeds than either A-S+ or A+S- scouts (Fig. 1m, red arrow). Differences in speed and directionality between wild type and A-S+ conditions were found statistically significant (Figs. S1i-j) and distributions were found robust to the classification parameters (Figs. S1k-n).”

“Secondly, we investigated the functional role of A-motility in the movement of swarms by quantifying their instantaneous speed distributions in wild-type and mutant communities. Remarkably, the removal of A-motility led to a large decrease in instantaneous swarm speeds (Figs. 3f and S3a). Interestingly, removal of S-motility only had an effect on the high instantaneous speeds of cells within swarms, suggesting that A-motility is sufficient to drive the movement of collective groups but may be synergistically enhanced when combined with S-motility. Differences in speed between wild type and A-S+ conditions were found statistically significant (Fig. S3b) and swarm speed distributions were found robust to the classification parameters (Figs. S3c-d). All in all, these results suggest that A-motile cells intermingle with S-motile cells and play a functional role in the collective movement of all cell populations. ”

2.3 Figure 2: *it would be nice to quantify the fraction of cells in each state for each mutant, and whether the possible differences are significant. This would strengthen the point that A-motile cells are more “exploratory”*

We thank the reviewer for their suggestion and have added the fraction of cells in each state for each condition (Fig. S1e).

Figure S1e : Box plot summaries of the fraction of motile cells for each class in wildtype (WT, blue boxes), S-motile (A-S+, orange boxes) and A-motile (A+S-, green boxes) communities. Each boxplot represents the statistics of single experiments. The corresponding statistical tests calculated using a t-test are $p=0.397$ between WT and A-S+ scouts, $p=0.043$ between WT and A+S- scouts; $p=0.608$ between WT and A-S+ loners, $p=0.011$ between WT and A+S- loners; and $p=0.647$ between WT and A-S+ swarms, $p=0.011$ between WT and A+S- swarms.

This new analysis is now referenced in the main text as:

“Loss of S-motility (A+S-) led to colonies with reduced swarm sizes at the invasion front (Fig. 1j, blue arrow and Fig. S1e), consistent with the classical result of Hodgkin⁸. As expected, scouts and loners were still present in this community (Fig. 1j, red and cyan boxes). In contrast, loss of A-motility (A-S+) produced communities with large swarms, but also with scouts and loners (Fig. 1k and Fig. S1a,e). This result seems in contrast to the classical view of A-motility being required to produce isolated single cells in non-predating conditions⁹. To understand this apparent discrepancy, we compared scout trajectories in wild-type and A-S+ communities. We observed that scouts in A-S+ colonies traveled considerably shorter distances than wild-type (Figs. 1l and S1f), which is expected since Type IV pili allow movement of cells only if attachment prey cells are within range^{21,22}. Only comparing populations of motile cells yielded classes consistent with the view that A-motility is associated with exploratory behaviors of single cells (Fig. S1e). ”

2.4 *Figure 4, Slime trail: It is very likely that the slime trail is responsible for the guiding effect of A-cells. Is this behavior lost in a non-secreting (or reduced secretion) mutant, or with appropriate surface treatment? I feel this is fairly easy to address and would strengthen Fig. 4 results.*

We appreciate the reviewer's suggestion regarding the role of slime trails in the guiding effect of A-cells and the possibility of investigating its influence through non-secreting or reduced secretion mutants or surface treatments. However, addressing this aspect is not a trivial task due to several factors.

Firstly, while it is likely that slime trails contribute to the guiding behavior of A-cells, the composition of the slime and the specific components recognized by the cells remain poorly characterized. Investigating the exact composition and recognition mechanisms of the slime would require extensive additional research and characterization.

Secondly, the prior study by Ducret et al. (Ducret et al. 2012) indicates that slime mediates cell adhesion to the substrate and is deposited by the A-motility complexes during motility. Therefore, mutating or reducing slime secretion could have profound effects on the motility of the A-motile cells, making it challenging to isolate the specific impact on guiding behavior without confounding factors.

Considering these limitations, we believe that addressing the influence of slime trails on guiding behavior would require a separate and more comprehensive study specifically focused on slime composition, recognition mechanisms, and their role in guiding A-cells. While we acknowledge the potential relevance of this aspect, the scope of our current study was primarily centered on exploring the overall behavior and classification of cells based on motility engines.

2.5 *Fig 5: it seems that most of the killing is done where the swarm is located. However, it could also be possible that the time it takes for a cell to lyse masks the effectivity of the scouts in killing. Was this time measured and how does it compare with the speed of the swarm front?*

We appreciate the reviewer's observation regarding the potential influence of cell lysis time on the effectiveness of scouts in killing. In fact, Seef et al. (Seef et al. 2021) previously quantified the time it takes for the prey to lyse upon *Myxococcus* contact. The authors consistently reported *E. coli* plasmolysis occurs within minutes, which is shorter than the time swarms need to reach the same area given their speed of motion ($\sim 2\mu\text{m}/\text{min}$). It is therefore unlikely that the prey killing we quantify is only due to scouts killing. Also, please note that in our article, we do not claim that swarms are solely responsible for the killing, but rather that they contribute to enhancing predation.

Secondary points:

2.6 *It is strange to me that the Voronoi area-cluster size histograms are not partially discretized along the x-axis. This suggests that there are non-integer numbers of cells per cluster. Please clarify how the histogram is computed to explain this.*

We apologize for the lack of clarity regarding the Voronoi area-cluster size histograms. The histograms were computed using a method where the number of cells in a cluster is determined by dividing the total area of the cluster by the average area of a single cell. This approach was chosen to mitigate errors in single-cell segmentation, particularly in cases involving large groups of cells densely packed together. However, due to the heterogeneity in cell size, non-integer values occasionally arise for the number of cells per cluster, resulting in the absence of partial discretization along the x-axis of the histograms. In our revised version, we have explicitly described our method to calculate histograms (Methods section) and highlighted the reasons behind the non-integer values observed.

The Long-range clustering method section now reads:

“For each identified cluster, its size was measured as the area covered by the non-dilated cell masks comprising each cluster and the number of cells per cluster was determined by dividing the total area of the cluster by the average area of a single cell. This approach was chosen to mitigate errors in single-cell segmentation, particularly in cases involving large groups of cells densely packed together. However, due to the heterogeneity in cell size, non-integer values occasionally arise for the number of cells per cluster, resulting in the absence of partial discretization along the x-axis of the histograms of Figs. 1f, 1j, 1k, 3d, 3e.”

2.7 Fig 1l: what is the red arrow?

Thank you for bringing this to our attention. The red arrow does not hold any significant meaning or contribute to the interpretation of the results. It is an unintentional residue from a previous version of the figure that was overlooked during the revision process. We have now removed the arrow from the figure.

2.8 Fig 2B / Fig S2 : it would be good to plot the 3 histograms on the same graph for easier comparison.

We appreciate the reviewer's suggestion to plot the three histograms on the same graph for easier comparison in Figs. 2b and S2. We have taken this suggestion into consideration and included the requested plot in supplementary Figure S2c.

Fig. S2c: Normalized histograms of the probability of the number of transitions per trajectory for wild-type, S-motile cells (A-S-) and A-motile (A-i-S-) cells data accumulated from six experimental replicates for the wildtype and four replicates for each mutant strains.

We have now added this new analysis and referenced it in the main text as:

“Importantly, state transitions were also frequent and symmetric in communities lacking A- or S-motility (Figs. 2d-e and S2a-c), thus the ability of cells to transition back and forth between states did not seem to depend on either motility system.”

2.9 Fig 4E and Fig S4G: Plotting the distributions in log scale along the y-axis would make the data more readable.

We thank the reviewer for this suggestion. We have implemented the requested modifications in the figures to improve the clarity and interpretation of the data for readers.

Revised Fig. 4e: Histogram of the length of overlapping tracks (Similarity track length) for wildtype, A-S+ and A+S-. Shaded areas highlight the standard deviations from the mean (solid line) of six experimental replicates for the wild-type and four replicates for each mutant strain.

Revised Fig. S4d: Histogram of the length of overlapping tracks (Similarity track size) used in the main figure for wild-type, S-motile cells (A-S+) and A-motile (A+S-) cells.

2.10 Figure S5B & D the frame seems to be the same at all time points.

These frames may give the impression of being identical, but they are in fact different. We understand that this perception can arise due to the impaired ability of the *Myxococcus* motility mutant strains to effectively prey on *E. coli*, resulting in stationary prey cells that are not lysed by *M. xanthus*.

To address this issue, we revised these figures by adding zoomed-in sections that allow for better observation of subtle differences. By examining these zoomed-in regions, one can notice slight variations that indicate changes occurring over time.

Revised Figs. S5a-d : (S5a) Evolution of predation over time at the predation forefront of S-motile cells (A-S+) visualized with semantically segmented large ROI containing the masks for *M. xanthus* (green) and *E. coli* (white). Two zones were defined: i) where no invasion occurred (orange shaded area) and ii) where *M. xanthus* cells invaded *E. coli* (blue shaded area). Scalebars = 100 pm. Scalebars of zoomed images = 20 pm. (S5b) Evolution of the total raw fluorescence signal from *E. coli* cells over time invaded by S-motile *M. xanthus* cells (A-S+). Orange and blue boxed areas highlight the safe zone, where no predation occurred and the predation zone, where active predation is occurring, respectively. Scalebars = 100 pm. Scalebars of zoomed images = 20 pm. (S5c-d) Same as panels A and B but for A-motile *M. xanthus* cells (A+S-)

2.11 experimental methods: please provide the magnification and NA of the objective, as well as the exact pixel size of the camera.

We now provide the requested information in the *Microscopy/Fast time-lapse imaging* section of the revised manuscript:

“A region of interest spanning an area of approximately 0.36 mm² was imaged through a 60X, 1.2NA objective by constructing a mosaic patchwork of 3 by 3 fields of view (FOVs), each of 2048x2048 pixels, overlapping by 200 pixels (1 pixel corresponding to 106 nm at the sample plane).”

2.12 methods of classification: the definition of V is ambiguous, I assume it is the Voronoi area and not the cell density (ie its inverse). The definition of the scouts seems to have a typo.

We thank the reviewer for pointing out the issue with the definition of "V" and the typographical error in the description of scouts. We have addressed and corrected these issues in the revised manuscript.

2.13 The "Figures" codes are not available on Github yet

We have now made the code for the Figures public.

2.14 *several references, eg Refs 1, 7, 15, 17 are incomplete*

We apologize and have fixed these referencing errors.

REVIEWERS' COMMENTS

Reviewer #1 (Remarks to the Author):

In the revised manuscript, the authors have addressed my comments appropriately. I think the work in the current version is publishable in Nature Communications. There are a couple of additional comments listed below regarding the revised manuscript, which the authors may consider to further improve the clarity of the manuscript.

For fig S1k-n, it will be better to plot the three curves in one graph in order to see whether and how the differences among curves are. In addition, it might be better to explicitly show how many cells are changed for each class when different threshold values are used. This information can be added either in corresponding figure captions or in the main text.

Similar for Figs. S3c-d, S4j-k.

Reviewer #2 (Remarks to the Author):

The authors have addressed most of our comments satisfactorily in my opinion. I do have a few small points to make though:

The only important remaining point for me that needs to be addressed is to provide methods for the UMAP projection, i.e. most importantly, which parameter space was projected, but also which parameters were used for projection.

Other small points:

- in Fig S1e, the legend indicates: "Each boxplot represents the statistics of single experiments." My understanding from the response to reviewer was that this was a quantification across replicates. Did I misunderstand something?

- Legend of Fig 1i the magnified image is on the bottom right

- Fig 1l : To be honest, this figure is less convincing than the plot of radii of gyration in showing the less exploratory behavior of A-S+, I would recommend swapping the figures.

REVIEWERS' COMMENTS

Reviewer #1

In the revised manuscript, the authors have addressed my comments appropriately. I think the work in the current version is publishable in Nature Communications. There are a couple of additional comments listed below regarding the revised manuscript, which the authors may consider to further improve the clarity of the manuscript.

We thank the reviewer for the positive comments.

For fig S1k-n, it will be better to plot the three curves in one graph in order to see whether and how the differences among curves are.

We understand that overlapping of the different curves may improve the comparison of differences, however, we are worried that a single plot with all the curves may considerably complicate the visualization of each individual dataset. To find a middle ground, we now provide visual cues that are positioned at the same location in each plot to simplify visualization of differences between datasets.

In addition, it might be better to explicitly show how many cells are changed for each class when different threshold values are used. This information can be added either in corresponding figure captions or in the main text. Similar for Figs. S3c-d, S4j-k.

For panels S1k-n, S3c-d, S4j-k, we now provide the number of cells for each class analyzed as insets for each plot, as the reviewer requests.

Reviewer #2

The authors have addressed most of our comments satisfactorily in my opinion. I do have a few small points to make though:

The only important remaining point for me that needs to be addressed is to provide methods for the UMAP projection, i.e. most importantly, which parameter space was projected, but also which parameters were used for projection.

We apologize for the lack of detail in the methods used for the UMAP projection. We have now included a dedicated section in the Methods describing how we applied UMAP embeddings, and what parameters were projected. In addition, we uploaded the actual embedding to our online OSF repository so readers could embed their data into this exact embedding. The new method section reads:

“UMAP Projection

Uniform Manifold Approximation and Projection (UMAP) was employed for dimensionality reduction and visualization³². A normalization step was performed on the input local density and cluster size features using the StandardScaler from scikit-learn. The UMAP algorithm was executed using a Number of Neighbors of 100 and a Minimum Distance of 0. UMAP's default values were used for other parameters, such as the distance metric (Euclidean) and the number of dimensions in the projected space (2D visualization). The UMAP projection results were visualized using Datashader. The embedding is provided at: <https://osf.io/kc3sw/>”

Other small points:

- in Fig S1e, the legend indicates: “Each boxplot represents the statistics of single experiments.” My understanding from the response to reviewer was that this was a quantification across replicates. Did I misunderstand something?

We revised the text to disambiguate. The new text reads:

Each boxplot represents the statistics across experimental replicates.

Legend of Fig 1 the magnified image is on the bottom right

We modified the legend to point to the right panel.

Fig 1l: To be honest, this figure is less convincing than the plot of radii of gyration in showing the less exploratory behavior of A-S+, I would recommend swapping the figures.

We followed the recommendation of the reviewer and swapped the figure panels.